# Dietary lipid content modifies *wah-1/AIFM1*-associated phenotypes via LRK-1 and DRP-1 expression in *C. elegans*

Mrityunjoy Mondal[1], Enzo Scifo [1], Rossella Erminia Ciliberti [1], Lena Wischhof[1], Tannaz Norizadeh Abbariki [1], Joshua Jackson [1], Ioanna-Maria Menegatou[1], Viktoria Zeisler-Diehl[2], Jan Riemer[3], Benjamin Jussila [4], Christopher E. Hopkins[4], Sylwia Kierszniowska [5], Lukas Schreiber [2], Pierluigi Nicotera[1,6], Dan Ehninger [1] & Daniele Bano [1] ✉

Eukaryotic cells rely on mitochondria to fine-tune their metabolism in response to environmental and nutritional changes. However, how mitochondria adapt to nutrient availability and how diets impact mitochondrial disease progression, remain unclear. Here, we show that lipid-derived diets influence the survival of *Caenorhabditis elegans* carrying a hypomorphic *wah-1/AIFM1* mutation that compromises mitochondrial Complex I assembly. Comparative proteomic and lipidomic analyses reveal that the overall metabolic profile of *wah-1/AIFM1* mutants varies with bacterial diet. Specifically, high-lipid diets extend lifespan by promoting mitochondrial network maintenance and lipid accumulation, whereas low-lipid diets shorten animal survival via overactivation of LRK-1 and DRP-1. We demonstrate that LRK-1 inhibition downregulates DRP-1 expression, reduces mitochondrial network fragmentation, and attenuates excessive autophagy, thereby rescuing the survival defects of *wah-1* mutants maintained on low-lipid diets. Together, these findings suggest that nutrition, and particularly lipid intake, may ameliorate certain disease phenotypes associated with an inherited mutation that disrupts mitochondrial bioenergetics.

Eukaryotic cells are highly dependent on mitochondria for a variety of biosynthetic processes, including the aerobic conversion of carbon substrates into energy in the form of ATP molecules. The transfer of electrons from reduced coenzymes to molecular oxygen occurs via the electron transport chain (ETC), with the respiratory complexes generating an electrochemical gradient across the inner mitochondrial membrane (IMM). The flow of protons through the Complex V/ATP synthase sustains ATP production from inorganic phosphate and ADP[1,2]. Since mitochondrial activity shapes a variety of cellular processes, inherited or environmentally driven mitochondrial lesions may increase the vulnerability of energy-demanding cell types (e.g., neurons and muscle cells), thereby leading to a wide spectrum of conditions that span from frailty and physical disability to severe metabolic syndromes and progressive neurodegenerative disorders[1,3–6]. In this regard, rare pathogenic variants are known to cause clinically distinct disease subtypes or syndromes, complicating the development of effective treatments. One example of this complex scenario involves the gene encoding

[1]German Center for Neurodegenerative Diseases (DZNE), Bonn, Germany. [2]Institute of Cellular and Molecular Botany (IZMB), University of Bonn, Bonn, Germany. [3]Institute for Biochemistry and Cologne Excellence Cluster on Cellular Stress Responses in Aging-Associated Diseases (CECAD), University of Cologne, Cologne, Germany. [4]InVivo Biosystems, Eugene, OR, USA. [5]metaSysX GmbH, Am Mühlenberg 11, Postdam-Golm, Germany. [6]IRCCS Mondino Foundation, via Mondino 2, Pavia, Italy. ✉e-mail: daniele.bano@dzne.de

mitochondrial apoptosis-inducing factor 1 (AIFM1, hereinafter referred to as AIF).

AIF is a membrane-tethered NADH- and FAD-containing oxidor-eductase primarily, although not exclusively, localized within the intermembrane space (IMS) of the mitochondria[7,8]. A large number of in vitro and in vivo studies demonstrated that AIF contributes to ETC biogenesis and oxidative phosphorylation (OXPHOS)[9–15]. Mechanistically, AIF binds coiled-coil-helix-coiled-coil-helix domain-containing protein 4 (CHCHD4)/MIA40 and stimulates the oxidative folding of IMS-imported polypeptides, some of which are ETC components[16–23]. Additional AIF interactors include cristae organizers (MICOS complex subunits) and adenylate kinase 2 (AK2), an enzyme that regulates ATP/ADP/AMP levels in the mitochondrial IMS[24,25]. Over the last fifteen years, several rare inherited forms of mitochondrial diseases have been traced back to AIFM1 mutations[7,8]. The first described genetic lesion within the AIFM1 gene was an in-frame tri-nucleotide deletion resulting in an AIF variant lacking the residue arginine 201 (p.Arg201del)[26]. The young infants carrying the AIFM1(p.Arg201del) mutation developed severe neurological symptoms and muscle wasting that were accompanied by clear metabolic signatures of aberrant mitochondrial OXPHOS[26]. Thus far, in vitro evidence of some AIF variants suggests that they are likely loss-of-function and structurally unstable proteins[26–31]. However, while the majority of patients carrying AIFM1 mutations exhibit clear signatures of impaired mitochondrial bioenergetics and other metabolic defects (e.g., lactic acidosis), clinical outcomes greatly vary among patients with regard to phenotypic expressivity and tissue specificity[8]. One explanation is that pathogenic traits may depend on variable penetrance of genetic modifiers and/or compensatory processes, as in other mitochondrial diseases[6]. However, while AIF's role in ETC biogenesis is certain, it is currently premature to rule out that the variable phenotypic outcomes associated with AIF deficiency may also depend on environmental cues. If so, environmental cues may represent an extra layer of challenges to mutation carriers at high risk of disease. Despite the obvious difficulties, a better understanding of genetic and environmental factors influencing the phenotypic features of disease variants would provide a unique opportunity to facilitate the development of effective interventions that are currently lacking[3,5].

The bacterivore nematode Caenorhabditis elegans naturally grows and feeds on variable microbial communities that form its microbiome and provide the large majority of its nutrients. To standardize conditions and increase experimental reproducibility, C. elegans is usually grown and maintained on lawns of monoxenic cultures, such as Escherichia coli B-type strain OP50[32] or BL21-derived HT115(DE3), the latter primarily used for RNAi experiments[33]. By growing wild-type (wt) C. elegans on E. coli HT115 strain, we previously showed that wah-1/AIFM1 downregulation inhibits mitochondrial respiration and shortens animal lifespan[34]. However, when we accidentally downregulated wah-1 using dsRNA-expressing OP50 instead of HT115 bacteria, we observed an increase in longevity, rather than a lifespan reduction, despite a similar impact on mitochondrial OXPHOS. Motivated by these initial observations, we reasoned that the nutritional status might shape pathological processes in a genetically tractable model organism carrying a pathogenic AIF variant, thereby defining the eventual relevance of diet-derived metabolites in disease etiology. Toward this end, we generated and characterized a C. elegans strain carrying a hypomorphic wah-1/AIFM1 mutation as an inherited genetic lesion causing phenotypic variability. We simplified the concept of different dietary regimes in the form of unique bacterial strains, since bacteria are the predominant source of biomolecules and micronutrients (e.g., vitamins) to the host. We herein report that diet-derived lipids modify the phenotypic outcomes of WAH-1/AIF deficiency, with lipid-rich dietary conditions stimulating C. elegans survival despite the defects in mitochondrial bioenergetics.

## Results

### Different bacterial strains influence the survival of wah-1 deficient C. elegans

To test the influence of diets on phenotypes linked to wah-1 deficiency, we employed OP50 and HT115 bacterial strains, as they provide different ratios of lipids, such as diacylglycerols (DAGs), triacylglycerols (TAGs), saccharides (Fig. 1A and Supplementary data 1), and other organic compounds as previously described[35–37]. Using a standard RNAi feeding protocol from hatching, we achieved comparable wah-1 downregulation levels in animals exposed to either OP50 or HT115 bacteria-derived double-stranded RNA (Supplementary Fig. S1A). In line with our previous data[34], wah-1 deficient nematodes had a shorter lifespan compared to controls when grown on HT115 bacteria, whereas wah-1 RNAi-competent OP50 induced animal longevity (Fig. 1B, C and Supplementary data 7). To confirm our observations in a model that recapitulates features associated with pathogenic WAH-1/AIF variants, we aimed to carry out gene editing to obtain a viable wah-1 hypomorphic C. elegans strain, rather than a wah-1 loss of function mutation that is known to cause embryonic lethality[38,39]. Since the catalytic domains of AIF and WAH-1 are structurally conserved (Supplementary Fig. S1B), we generated a WAH-1 variant lacking 9 amino acids (WAH-1(p.GKKRDIFYE deletion)) within the putative FAD-binding domain and around the R309 corresponding to the disease-causing AIF mutation (p.Arg201del) identified in humans[26] (Fig. 1D). Among the most notable phenotypes, hypomorphic wah-1(bon89) mutants exhibited developmental delay at 20 °C and larval arrest at 25 °C (Supplementary Fig. S1C, D). Moreover, adult wah-1(bon89) mutant hermaphrodites had a higher number of engulfed apoptotic corpses in the gonads and produced a smaller brood size compared to controls (Supplementary Fig. S1E, F). Next, we profiled the proteome of wah-1(bon89) mutants and found a consistent reduction of mitochondrial Complex I subunits (NDUB-2, GAS-1, NDUV-2, NDUB-7, NDUO-1, NDUA-12, NDUA-5, NDUF-7, NDUS-8) compared to wild types (Fig. 1E and Supplementary data 2). The loss of a Complex I subunit was confirmed by immunoblots and was associated with a lower oxygen consumption rate (Fig. 1F, G). Additional prominent proteomic signatures included the upregulation of enzymes involved in glucose metabolism (ALDO-1, GPD-3, ENOL-1, PYC-1, LDH-1, MDH-2), mitochondrial stress (GST-4, HSP-6), ubiquinone synthesis (CLK-1) and intermediates of the tricarboxylic acid cycle (SDHD-1, SDHA-1, ACO-2, MMAA-1), whereas fatty acid desaturase FAT-7 was significantly downregulated (Fig. 1E and Supplementary Fig. S1G). Furthermore, wah-1 mutants showed an upregulation of proteins involved in mitochondrial protein import (TOMM-70, TIN-44), chaperones (HSP-6, DNJ-21, DNJ-10), and quality control (CLPP-1, SPG-7, PHB-1) (Supplementary Fig. S1H). Consistent with the role of AIF/CHCHD4 complex in the oxidative folding of mitochondrial precursors[8,18], several bona fide substrates were downregulated in wah-1 mutant nematodes (Fig. 1E, Supplementary Fig. S1H and Supplementary data 2), including LET-754/AK2[24,25], FAMH-136/FAM136A[15,40], TIN-13/TIMM13, DDP-1/TIMM8A, NDUB-7/NDUFB7 and NDUFS-8/NDUFS8[18]. In summary, hypomorphic wah-1(bon89) animals showed molecular signatures similar to those observed in patients and AIF-deficient mice[8,18,19].

Next, we carried out lifespan assays on OP50 and HT115 bacteria. Consistent with our RNAi data (Fig. 1B, C), wah-1(bon89) mutant nematodes lived significantly longer (approximately 2 days) than wt animals on OP50, whereas they had a shorter lifespan (approximately 4 days) than wt animals when grown on HT115 bacteria (Fig. 1H, I and Supplementary data 7). To test if different bacterial diets could impact WAH-1 stability and/or expression, we performed SureQuant-based targeted mass spectrometry using two validated WAH-1 unique peptides (FAPHLHINAIGK and YPAEDILPE-HIAQK) corresponding to WAH-1 residues 602-613 and 447-460, respectively (Fig. 1J, K). WAH-1 protein levels were comparable

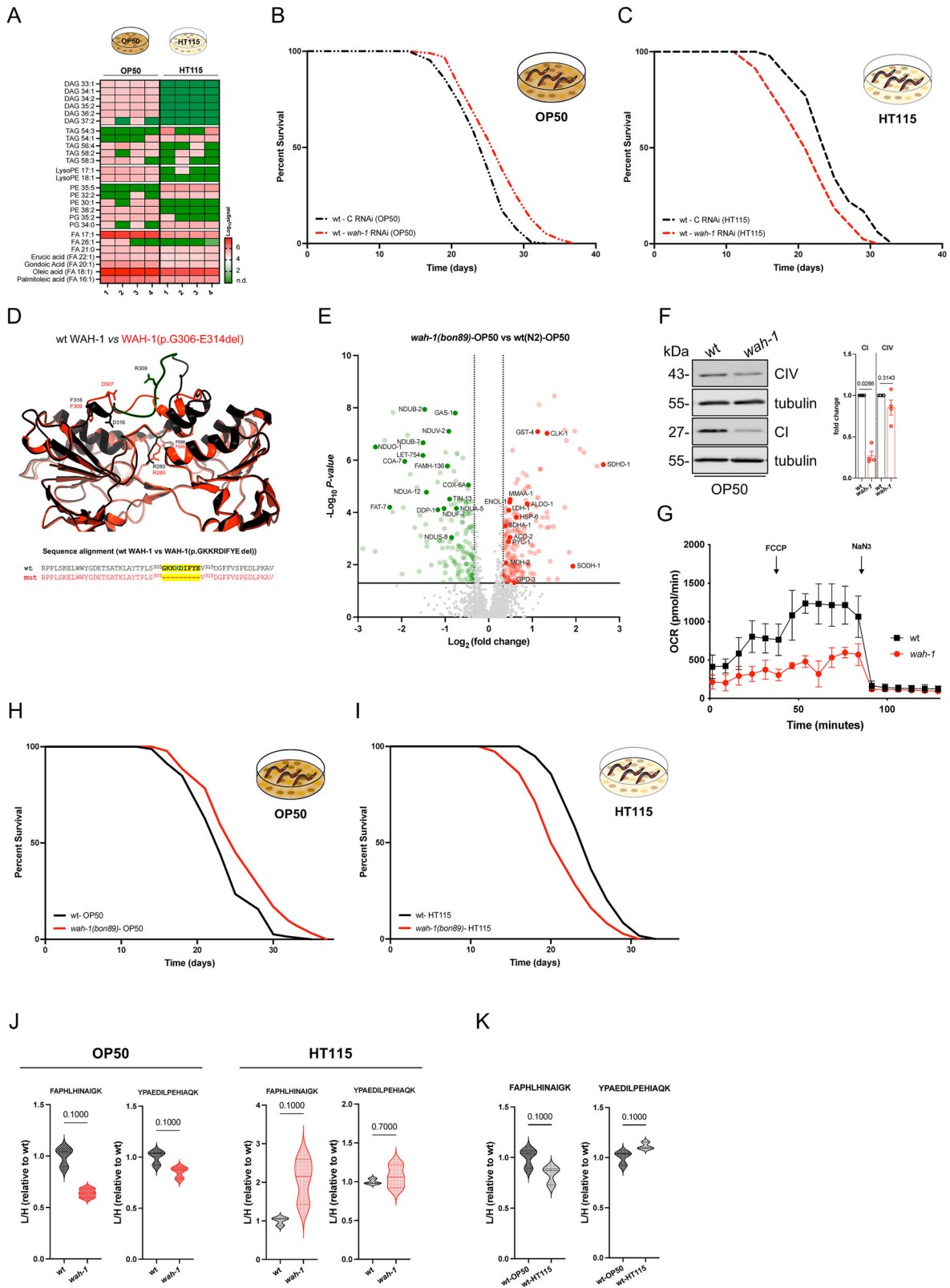

between wt and *wah-1(bon89)* animals, with a trend, although not significant, toward a lower amount in *wah-1* mutants fed with OP50 bacteria (Fig. 1J). Thus, changes in WAH-1 protein expression and/or stability are likely not the main molecular consequences of bacterial diets. Taken together, our data suggest that diet-derived nutrients may influence the phenotypic outcome of a pathogenic WAH-1/AIF variant.

## Diets differentially regulate SKN-1 and mitochondrial network homeostasis

Inhibition of mitochondrial bioenergetics can extend *C. elegans* lifespan through "mitohormesis"[41], a transcriptionally regulated cellular response that occurs during development and has long-lasting protective adaptations. Since proteins associated with "mitohormesis" (HSP-6 and GST-4) were upregulated in our proteomics (Fig. 1E), we

**Fig. 1 | WAH-1 deficiency inhibits Complex I and affects *C. elegans* survival.**
**A** Heat map of detected lipids in *E. coli*-type strain OP50 and K-12 strain HT115(DE3). For each condition, 4 biological replicates (numbered from 1 to 4) were used and analyzed. Colors indicated $\log_{10}$ signal (n.d.=not detected, as signal below the detection threshold). Representative survival curves of wt (N2) animals grown either on *E. coli* **B** OP50 or **C** HT115 strains expressing control (black lines) and *wah-1* (red lines) RNAi. **D** AlphaFold2 (AF2) prediction and structure alignment of FAD-containing and C-terminal domains (Tyr242-Thr600) of wt (black) and mutated (red) WAH-1 variants (Uniprot ID/accession number of wt WAH-1 = Q9U229). The deleted sequence (Gly305-Glu313) is highlighted in green. At the bottom, the predicted WAH-1/AIF protein sequence lacking 9 amino acids (mut, in red) is based on DNA sequencing. **E** Volcano plot of dysregulated proteins in *wah-1(bon89)* compared to wt nematodes (OP50 bacteria). Two-tailed t-test was used for statistical analysis. Thresholds: *p*-value- $\leq 0.05$ and FC- $\pm 1.25$. Highlighted are some up- (red) and down- (green) regulated proteins involved in mitochondrial bioenergetics and carbon metabolism. **F** Representative immunoblots of samples from wt and *wah-1* mutants using validated antibodies against NUO-2/NDUFS3 (CI), CTC-1/MT-CO1 (CIV), and tubulin. Densitometry is reported on the right ($n = 4$, Mean +/− SEM, two-tailed test, Mann-Whitney test). **G** Representative Seahorse experiment of wt and *wah-1* mutant nematodes. Oxygen consumption rate (OCR) was measured over time and upon exposure to FCCP and $NaN_3$ (Mean +/− SD, $n = 5$) Representative survival curves of wt (N2) and *wah-1(bon89)* mutant animals grown either on *E. coli* (**H**) OP50 or **I** HT115. **J, K** SureQuant-based MS using two specific WAH-1 peptides (FAPHLHINAIGK and YPAEDILPEHIAQK) for spiking the samples. Graphs report light/heavy ratios (L/H) of WAH-1 expression in wt and *wah-1* mutant nematodes grown on *E. coli* OP50 and HT115 strains. In **K**, comparison of WAH-1 protein expression in wt *C. elegans* grown either on OP50 or HT115 bacteria ($n = 3$; two-tailed t-test, Mann-Whitney test).

tested whether the two diets could differentially influence mitochondrial unfolded protein response (UPR^mt) and antioxidant pathways. To do so, we employed transgenes encoding GFP under *hsp-6*, *hsp-60*, *dve-1* promoters as well-established markers of UPR^mt[42], while a [*gst-4p::GFP::NLS*] transgene was used to assess SKN-1/Nrf-dependent detoxification pathways[43,44]. In both OP50 and HT115, *wah-1(bon89)* showed an upregulation of [*hsp-6p::GFP*], [*hsp-60p::GFP*], and [*dve-1p::dve-1::GFP*] compared to controls (Fig. 2A, B), further confirming the UPR^mt induction. However, ATFS-1 loss, a major transcriptional regulator of UPR^mt[42], had no impact on *wah-1(bon89)* survival on OP50 (Fig. 2C and Supplementary data 7). Interestingly, *wah-1(bon89)* mutants showed a stronger upregulation of [*gst-4p::GFP::NLS*] compared to controls when grown on OP50 rather than HT115 (Fig. 2A, B). Contrary to *atfs-1* deficiency, OP50-expressing dsRNA against *skn-1* negatively affected *wah-1* survival and mitochondrial morphology (Fig. 2D and Supplementary Fig. S2A), suggesting a causative role of SKN-1/Nrf in diet-dependent lifespan extension. Of note, the insulin/IGF-1/DAF-2 signaling pathway had little to no effect on the survival of *wah-1* mutants (Supplementary Fig. S2B and Supplementary data 7).

To gain further mechanistic insights, we analyzed the proteomic profiles of *wah-1(bon89)* mutant animals on HT115. Consistent with the previous signatures in animals grown on OP50 (Fig. 1E and Supplementary Fig. S1H), HT115-fed *wah-1* mutants showed decreased expression levels of known AIF/CHCHD4-binding substrates (Supplementary Fig. S2C and Supplementary data 3). When we compared the proteomic changes of *wah-1* mutants grown on the two different bacterial strains, we observed many differentially regulated mitochondrial proteins, including DRP-1, STL-1, CHCH-3, and ATAD-3, which are known regulators of mitochondrial membrane remodeling (Fig. 2E, F and Supplementary data 4). Using strains expressing *drp-1* endogenously tagged with GFP, we confirmed DRP-1::GFP upregulation in *wah-1(bon89)* mutants on HT115 compared to OP50 bacteria (Fig. 2G). Consistently, body wall muscle cells expressing a mitochondrial matrix-targeted GFP exhibited a more fragmented mitochondrial network when *wah-1(bon89)* animals were grown on HT115 bacteria instead of the OP50 diet (Fig. 2H). Furthermore, *wah-1* mutants tend to have higher mitochondrial DNA (mtDNA) contents as measured by MT-CO1 and MT-ND1 mRNA levels, indicating an increased mitochondrial mass (Supplementary Fig. S2D). We assessed the energy status of our animals by employing the ratiometric biosensor Queen-2m, a GFP-based indicator that undergoes conformational changes upon ATP binding, which enables the quantification of ATP dynamics in living cells[45,46]. On OP50 but not on HT115, *wah-1(bon89)* animals had higher Queen-2m signals compared to controls (Fig. 2I, J), likely as a result of higher cytosolic ATP levels. Finally, we assessed mitochondrial morphology in control and AIF KO HEK293T cells and found that oleic acid supplementation was sufficient to promote a more fused mitochondrial network (Supplementary Fig. S2E). Also, oleic acid could rescue the survival defects of glucose-deprived, galactose-fed AIF KO HEK293T cells (Supplementary Fig. S2F), further supporting the idea that lipids can ameliorate defects associated with AIF deficiency.

## Diets influence storage lipids and energy homeostasis in *wah-1* mutants

Since storage lipids are particularly abundant in *C. elegans* gut[36,47,48], we assessed lipid droplets (LDs) with the fusion reporter DHS-3::GFP[49,50]. We quantified a lower number of DHS-3::GFP-containing lipid droplets in *wah-1* mutants compared to wt nematodes when grown on OP50 bacteria (Fig. 3A). However, nematodes had even fewer DHS-3::GFP-positive LDs on HT115 bacteria, with *wah-1* mutants having a few detectable ones (Fig. 3A). To obtain a broader and more quantitative analysis, we employed mass spectrometry-based lipidomics and found that *wah-1* mutants on OP50 had more TAGs and much less phosphatidylcholines (PCs) compared to HT115-fed counterparts (Fig. 3B, C and Supplementary data 5). To test if supplementation of a single mono-unsaturated fatty acid could rescue *wah-1* lifespan reduction, we fed nematodes with oleic acid and observed no changes in *wah-1* survival on HT115 (Supplementary Fig. S3A and Supplementary data 7). Likewise, other supplements (glucose and Vitamin B12) in OP50 bacteria had negligible effects on *wah-1* lifespan (Supplementary Fig. S3B and Supplementary data 7). We reasoned that a complex mixture of lipids is necessary to rescue lifespan defects; therefore, we expanded our experimental paradigms by using *E. coli* K-12 (BW25113) strains obtained from the Keio knockout library[51]. After an in silico screening of bacterial mutants with reported effects on lipids, we focused on diacylglycerol kinase (Δ*dgkA*) mutant strains (Supplementary Fig. S3C), knowing that loss of diacylglycerol kinase alpha negatively affects the conversion of DAGs into phosphatidic acid (PA)[52,53] and promotes TAG accumulation in *E. coli* mutants[54]. Metabolomic analyses confirmed higher lipid contents (e.g., DAGs, TAGs) in *E. coli* K-12 Δ*dgkA* compared to control bacteria (Fig. 3D and Supplementary Data 1). As in OP50, *wah-1* mutants on *E. coli* K-12 Δ*dgkA* had more TAGs than those fed on control *E. coli* K-12 (BW25113) (Fig. 3E, F, and Supplementary Data 6). The increased lipid content correlated with a higher number of DHS-3::GFP signals in both wt and *wah-1(bon89)* mutant nematodes on *E. coli* K-12 Δ*dgkA* (Fig. 3G). Importantly, hypomorphic *wah-1(bon89)* nematodes on *E. coli* K-12 Δ*dgkA* showed a more fused mitochondrial network, lower DRP-1::GFP expression and lifespan extension (Fig. 3H–J and Supplementary Data 7). On the contrary, *E. coli* K-12 control negatively affected LD number, mitochondrial morphology and survival of *wah-1* mutants compared to wt (Fig. 3G–I and Supplementary Data 7). Similarly to nematodes fed on OP50, *wah-1* mutants grown on *E. coli* K-12 Δ*dgkA* had higher cytosolic ATP levels than those on *E. coli* K-12 control (Fig. 3K). Although *E. coli* K-12 Δ*dgkA* had negligible effects on Complex I loss and UPR^mt induction, it also promoted the expression of the [*gst-4p::GFP::NLS*] transgene (Supplementary Fig. S3D–H), possibly indicating the activation of SKN-1/Nrf pathway. Together, our data suggest that TAG-rich diets modify molecular signatures and phenotypes linked to hypomorphic *wah-1/AIFM1* mutation.

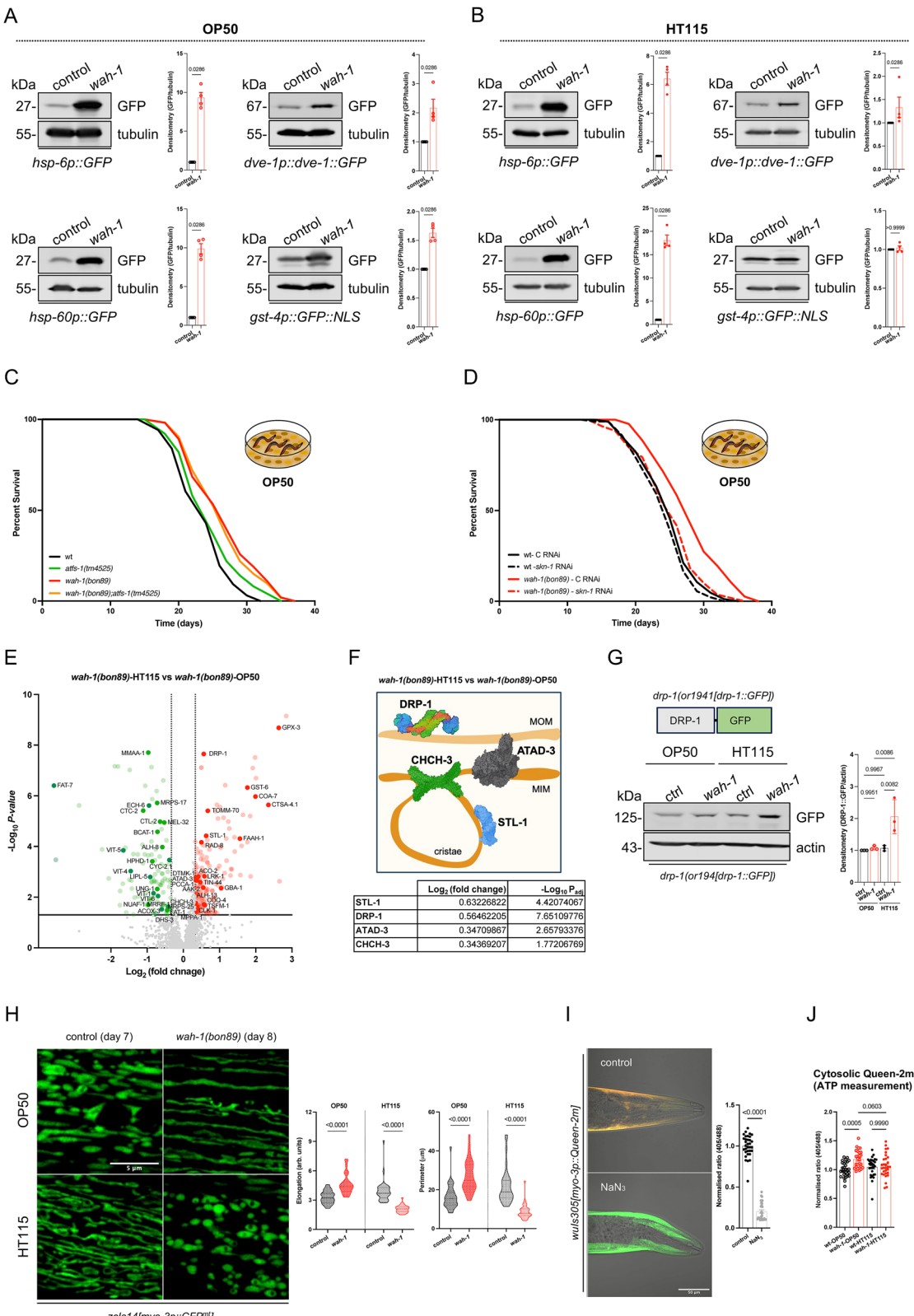

## LRK-1 and DRP-1 upregulation negatively affects the survival of *wah-1* mutants

We sought to explore the mechanisms by which low-lipid diets could influence the pathogenic outcome of a hypomorphic *wah-1* mutation. Upon closer examination of our comparative analyses (Fig. 2E, Supplementary Fig. S4A and Supplementary Data 4), we found that many mitochondrial proteins, as well as enzymes involved in lipid

metabolism, lipid transport, lysosomal and autophagic pathways were differentially regulated in *wah-1* mutants grown on different diets. We initially focused on upregulated mitochondrial proteins and performed an RNAi screen (HT115 bacteria) with *wah-1* mutants, followed by a counter-screening of positive hits in wt *C. elegans* (Fig. 4A, B and Supplementary data 8). Out of the four candidates (LRK-1, DRP-1, ACO-2, and MPPA-1) identified in our RNAi screening, only LRK-1 and DRP-1

**Fig. 2 | Bacterial strains differentially influence stress response in *wah-1* mutants.** GFP expression levels in controls and *wah-1(bon89)* mutants expressing the indicated transgenes and grown either on **A** OP50 or **B** HT115. Densitometry is reported on the right of each representative immunoblots (*n* = 4, Mean +/− SEM, two-tailed t-test, Mann-Whitney test). **C** Lifespan assay of wt (N2), *atfs-1(tm4525)*, *wah-1(bon89)* and *wah-1(bon89);atfs-1(tm4525)* mutants grown on OP50. **D** Representative lifespan assay of wt and *wah-1(bon89)* on control and *skn*-1 RNAi (OP50). **E** Volcano plot of dysregulated proteins in *wah-1(bon89)* animals on HT115 and compared to *wah-1(bon89)* mutants on OP50 (two-tailed t-test, thresholds are: *p* value- ≤0.05 and FC- ±1.25). Highlights are mitochondrial proteins and metabolic enzymes. **F** Scheme reports the localization of DRP-1 (MOM= mitochondrial outer membrane), ATAD-3 (with domains that span across the two membranes), CHCH-3 (part of the MICOS complex), STL-1 (in the MIM, binding cardiolipin at the cristae). Created in BioRender. Bano, D. (2026) https://BioRender.com/xbjr3z5. At the bottom, proteomic changes of the four proteins are reported (data from Fig. 2E). **G** Representative immunoblots of control (ctrl) and *wah-1* mutant animals expressing *drp-1(or1941[drp-1::GFP])* and grown on either OP50 or HT115. Densitometry is reported on the right (*n* = 3, Mean +/− SEM, one-way ANOVA, Šídák's multiple comparisons test). **H** Representative confocal images of the mitochondrial network in body wall muscle cells of control and *wah-1(bon89)* animals grown on OP50 or HT115 bacteria. On the right, quantification of mitochondrial elongation and perimeter (*n* = 30–35 animals/condition from 3 biological replicates, one-way ANOVA, Šídák's multiple comparisons test). **I** Representative confocal images of Queen-2m in the body wall muscle cells. Red (emission at 405 nm) and green (emission at 488 nm) channels were merged onto bright-field images (superimposed color green: low ATP; yellow: high ATP). The plot represents relative signal changes (correlating with ATP availability) upon NaN$_3$ treatment (*n* = 33 (control) or 29 (NaN$_3$) animals/condition from 3 biological replicates, two-tailed t-test, Mann-Whitney test). **J** Quantification of cytosolic Queen-2m fluorescence in wt and *wah-1* mutants grown on OP50 or HT115 (*n* = 30 or 27 (for *wah-1* on OP50) animals/condition from 3 biological replicates, one-way ANOVA, Šídák's multiple comparisons test).

downregulation stimulated mitochondrial elongation in *wah-1* mutant animals (Fig. 4C and Supplementary Fig. S4B), although there was no noticeable effect on LDs in *C. elegans* guts (Supplementary Fig. S4C–E). Importantly, downregulation of leucine-rich repeat kinase LRK-1 and dynamin-related protein DRP-1 rescued the shorter lifespan of *wah-1* mutants grown on HT115 bacteria, whereas it did not have any noticeable effect on wild types (Fig. 4D and Supplementary data 7).

Parkinson's disease (PD)-related LRRK2 and its paralog LRRK1 are poorly characterized multifunctional protein kinases involved in endosomal trafficking, autophagy-lysosomal pathway, mitochondrial dynamics (through DRP1), and oxidative stress[55]. Since our proteomics showed clear lysosomal and autophagic signatures (Supplementary Fig. S4A), we quantified autolysosomes in *C. elegans* strains expressing the *sqIs11[lgg-1p::mCherry::GFP::lgg-1]* transgene[56]. Compared to wt animals, *wah-1* mutants had more mCherry-positive autolysosomes when grown on HT115 (Fig. 4E). Interestingly, single RNAi against either *drp-1* or *lrk-1* inhibited autophagy in *wah-1* mutant nematodes on HT115 (Fig. 4F). It is known that LRRK2 and DRP1 contribute to mitochondrial network maintenance via autophagy/mitophagy[57–60]. Although we could not quantify mitochondria-autophagosome co-localization in *C. elegans* due to technical limitations, we sought to test the contribution of PINK1/Parkin-dependent mitophagy[61] (Fig. 4G). RNAi against *pink-1* partially rescued the survival defects of *wah-1* mutants on HT115 (Fig. 4H and Supplementary data 7), with *pink-1*-deficient animals showing a more fused mitochondrial network (Fig. 4I). These data suggest that enhanced autophagy and mitophagy may contribute to the lifespan reduction of hypomorphic *wah-1* animals grown on low lipid-diets.

To better elucidate the interplay between LRK-1 and DRP-1 in animals grown on HT115 bacteria, we generated *wah-1(bon89);drp-1(tm1108)* double mutants and found that *drp-1(lof)* rescued the lifespan reduction of *wah-1* mutants on HT115 bacteria (Fig. 5A and Supplementary Data 7). Conversely, DRP-1 overexpression had a negative impact on hypomorphic *wah-1* nematodes on OP50 bacteria (Fig. 5B-C and Supplementary Data 7). We carried out additional epistatic analyses and found that *lrk-1* RNAi did not have an additive effect on the survival of *wah-1(bon89);drp-1(tm1108)* (Fig. 5D and Supplementary Data 7). Interestingly, immunoblot analyses showed that DRP-1::GFP upregulation in HT115-fed *wah-1* nematodes was inhibited by *lrk-1* RNAi (Fig. 5E), suggesting that LRK-1 may act upstream of DRP-1.

It is known that pathogenic LRRK2 mutations (G2019S and I2020T) alter the kinase activity of LRRK2[55]. Since the kinase domain is relatively conserved between human LRRK2 and *C. elegans* LRK-1, we carried out gene editing and obtained two missense insertion mutations that resulted in truncated variants, likely loss of function (Fig. 5F). Lifespan assays showed that the two *lrk-1* mutant strains lived as long as wt (Fig. 5G and Supplementary data 7). Consistent with our previous RNAi data (Fig. 4D), both *lrk-1* mutations could promote lifespan

extension of *wah-1(bon89)* animals on HT115, whereas *drp-1* RNAi had no additional effect on the survival of *lrk-1;wah-1* double mutants (Fig. 5H and Supplementary data 7). Both *lrk-1* mutations had no effect on Complex I in *wah-1* mutants (Fig. 5I). Together, these mechanistic data suggest that LRK-1 and DRP-1 act in the same pathway and their activity contributes to the phenotypic outcomes of hypomorphic *wah-1* mutant animals grown on different diets (Fig. 5J).

## Discussion

Over the last decade, a large number of inherited pathogenic mutations within the *AIFM1* gene have been identified in patients[8]. As for other metabolic disorders associated with aberrant mitochondrial bioenergetics, pathogenic *AIFM1* mutations cause a large spectrum of clinical outcomes, including fatal encephalomyopathy, ataxia, late-onset neuromyopathy, and mild deafness[8]. This range of clinical manifestations may result from varying degrees of structural and functional defects in AIF variants. Alternatively, disease-causing AIF variants may bind molecular partners differently compared to wild-type AIF, as recently shown for CHCHD4 and adenylate kinase 2 (AK2)[24,25]. Moreover, currently unknown genetic modifiers may ameliorate the metabolic defects due to *AIFM1 loss of function* mutations. While some of these molecular factors have been considered and thoroughly investigated in experimental models, less is known about the environmental cues that may contribute to the phenotypic outcomes linked to AIF deficiency.

To shed light on this, we sought to model AIF deficiency in isogenic populations of *C. elegans* hermaphrodites. Having obtained a viable *C. elegans* strain expressing a relatively stable WAH-1/AIF variant lacking 9 amino acids in the FAD-containing domain, we carried out functional assays as well as in-depth proteomic analyses. We found that the expression of hypomorphic *wah-1(bon89)* mutation negatively affects substrates of the AIF/CHCHD4 complex, resulting in aberrant Complex I biogenesis and impaired mitochondrial respiration, as previously shown in higher organisms[12,20,21,23]. By leveraging the genetic tractability of *C. elegans*, we studied whether environmental cues could modulate pathogenic traits linked to a *wah-1/AIFM1* hypomorphic mutation. When we grew *C. elegans* on two standard bacterial types (OP50 and HT115) that are known to provide different nutrients to the host[35–37], we found that the lifespan of hypomorphic *wah*-1 mutants varies significantly, while there was no effect on wild-type animals. We confirmed these observations with alternative bacterial strains and concluded that lipid-rich diets stimulate the accumulation of storage lipids (TAGs) and extend the lifespan of *wah*-1 mutants. Instead of a single fatty acid or a bacterially derived micronutrient (e.g., vitamin B12), our current data suggest that pathogenic phenotypes can be modulated when animals are exposed to diets rich in diverse lipids, which are subsequently used for energy production, biosynthetic processes, and membrane

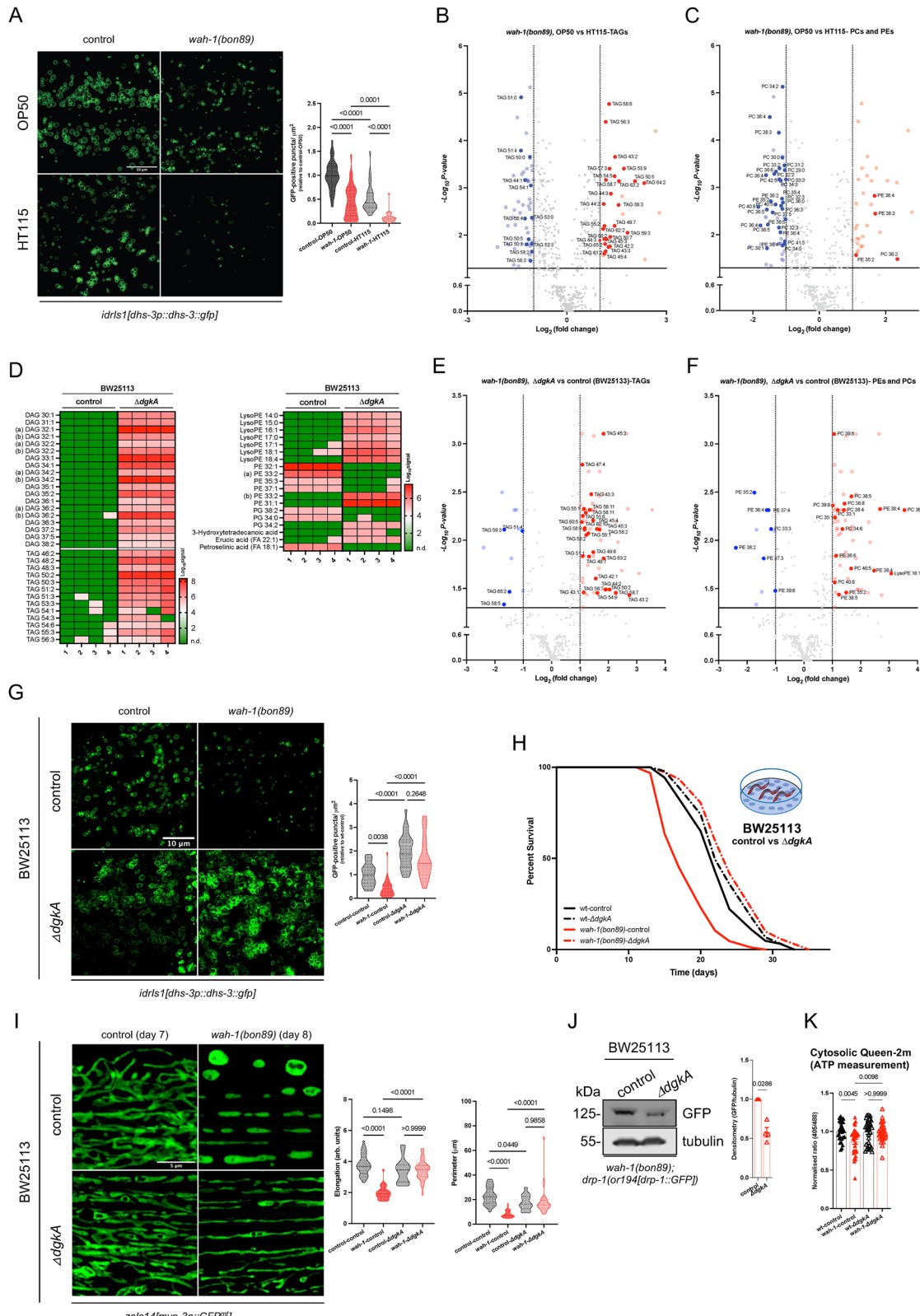

remodeling. As one of the main findings of our study, we were able to determine the molecular mechanisms by which diets can regulate the survival of animals expressing a hypomorphic WAH-1/AIF variant. Specifically, we found that low-lipid diets promote the expression of LRK-1 and DRP-1 to regulate mitochondrial network homeostasis in response to dietary conditions. By acting within the same pathway, inhibition of either DRP-1 or LRK-1 was sufficient to rescue several

molecular signatures in *wah-1* mutant animals grown on low-lipid diets, uncovering disease modifiers that are recruited in response to the environment. Taking a broader view, it is interesting to see how proteins often associated with PD are activated by OXPHOS deficiency to promote mitochondrial network maintenance. Since Complex I defects are usually observed in postmortem tissues from idiopathic PD patients[62], we speculate that some of the cellular

**Fig. 3 | Bacterial strains differentially affect storage lipids in *wah-1* mutant nematodes. A** Representative confocal images of control and *wah-1(bon89)* animals expressing DHS-3::GFP. Quantification of GFP signal is provided on the right (*n* = 30−36 animals/condition from 3 biological replicates, one-way ANOVA, Šídák's multiple comparisons test). Volcano plots of differentially regulated (**B**) triacylglycerols (TAGs) and (**C**) phosphatidylcholines (PCs) and phosphatidylethanolamines (PEs) in *wah-1* mutants grown on OP50 vs animals on HT115 (two-tailed t-test, thresholds are: *p* value- ≤ 0.05 and log$_2$FC- ±1). **D** Heat maps relative to the metabolomic analysis of *E. coli* K-12 control and *ΔdgkA* bacteria (BW25113 strain). For each condition, 4 biological replicates (indicated as 1 to 4) were analyzed. In the heatmap, (a) and (b) indicate lipid species with different chemical structure, but similar degrees of saturation. Data are reported in log$_{10}$ signal scale (n.d.= not detected, as signal below the detection threshold). Volcano plots of differentially regulated (**E**) TAGs and (**F**) PEs and PCs in *wah-1* mutants grown either on *E. coli* K-12 *ΔdgkA* or control (BW25113) (two-tailed t-test, thresholds are: *p* value- ≤ 0.05 and log$_2$FC- ±1). **G** Representative confocal images of DHS-3::GFP-positive lipid droplets

in control and *wah-1(bon89)* animals grown on *E. coli* K-12 control or *ΔdgkA* (BW25113). Quantification is provided on the right (*n* = 28−33 animals/conditions from 3 biological replicates, one-way ANOVA, Šídák's multiple comparisons test). **H** Representative survival curves of wt (N2) and *wah-1(bon89)* animals grown on *E. coli* K-12 control and *ΔdgkA*. **I** Representative confocal images of mitochondrial morphology in animals grown on *E. coli* K-12 control and *ΔdgkA*. Quantification reports elongation score and perimeter (*n* = 26-30 animals/condition from 3 biological replicates, one-way ANOVA, Šídák's multiple comparisons test). **J** Representative immunoblots of samples from *wah-1* mutants expressing *drp-1(or1941[drp-1::GFP])* grown on control and *ΔdgkA* bacteria (BW25113 strain). Densitometry is reported on the right (*n* = 4, Mean +/− SEM, two-tailed t-test, Mann-Whitney test). **K** Quantification of cytosolic Queen-2m fluorescence in wt and *wah-1* mutants grown on *E. coli* K-12 control and *ΔdgkA* (*n* = 30 animals/condition from 3 biological replicates, Mean +/− SEM, one-way ANOVA, Šídák's multiple comparisons test).

processes herein described might also contribute to the pathogenesis of PD.

As expected for an inherited pathogenic lesion, lipid-rich diets did not rescue Complex I defects and AIF/CHCHD4 dysfunction, suggesting that diet-derived lipids can stimulate stress response mechanisms that can compensate for the decreased mitochondrial respiration. In line with our previous studies[63–65], mobilization of lipid deposits can transform short-lived mitochondrial mutants into long-lived nematodes by inducing cellular adaptations that buffer metabolic defects and partially restore homeostasis. Consistent with this model, we showed that SKN-1/Nrf, but not ATFS-1, contributes to the lifespan extension of *wah-1* mutants on a lipid-rich bacterial diet. On the other hand, lower storage lipids in *wah-1* mutants correlated with decreased cytosolic ATP levels and a more fragmented mitochondrial network associated with higher expression of proteins involved in mitochondrial dynamics, including DRP-1. These cellular and molecular signatures were accompanied by enhanced autophagy and PINK1/Parkin-dependent processes (e.g., mitophagy) that negatively affected the survival of hypomorphic *wah-1* mutant animals on a low-lipid diet. Does it mean that autophagy and mitophagy are detrimental? Perturbations of the mitochondrial OXPHOS can induce transcriptional programs during development that have long-lasting effects on animal physiology. In experimental models such as *C. elegans*, loss of mitochondrial respiratory complexes triggers the production of reactive oxygen species (ROS), which can induce "mitohormesis" across tissues that enhances stress resilience and metabolic adaptations underlying longevity[41]. As cells grow during development, the expansion of the mitochondrial network supports the formation of contact sites between intracellular organelles, thereby facilitating the exchange of intermediate metabolites, ions, and lipids[64,66]. To buffer excessive ROS, the mitochondrial network undergoes continuous remodeling to ensure the degradation of damaged components (e.g., respiratory complexes, oxidized enzymes) via the formation of autophagosomes, for which mitochondria are the main suppliers of membranes[67]. However, in OXPHOS-deficient cells, the high turnover of mitochondria and the sustained flux of mitochondrial membranes toward other organelles may become increasingly difficult without an adequate dietary lipid intake to replenish those diverted to other cellular structures or used to generate ATP. Consequently, prolonged autophagy and mitophagy can undermine mitochondrial network homeostasis, potentially triggering vicious cycles of degradation and dysfunction that further impair the cell's bioenergetic capacity. Thus, while autophagy and mitophagy are essential homeostatic processes that protect cells from the accumulation of damaged cellular components, their excessive activation could be detrimental and causative of human diseases, as recently shown[68].

Given the current knowledge in the field and the limited number of clinical cases worldwide, it is difficult to translate our findings into

meaningful interventions for patients. We are aware of the limitations of our data, since they were mainly obtained in invertebrate models. Although more work is needed to draw definitive conclusions, our study provides robust proof-of-principle concepts of evolutionarily conserved molecular processes that likely may also be relevant in mammals. Furthermore, we believe that our findings increase the awareness of diets as disease modifiers, since complex nutrients may influence disease progression in an organism carrying a pathogenic mutation. In support of this claim, it was suggested that ketogenic diets might slow down mitochondrial myopathy progression in transgenic mice and improve muscle regeneration and strength in adult patients with progressive external ophthalmoplegia[69,70]. Similarly, a few clinical trials have been undertaken to investigate the impact of nutritional interventions for the treatment of progressive neurodegenerative diseases, such as amyotrophic lateral sclerosis (ALS), in which hypometabolism and aberrant mitochondrial bioenergetics play a causative role[71–73]. Consistent with a scenario applicable also to other metabolic disorders, there is a direct relationship between ALS risk and the levels of circulating lipids[74,75], further emphasizing that lipid availability may also modify the disease trajectory in humans.

In conclusion, our study reveals that diet-genotype-phenotype correlations are overlooked aspects of pathogenic processes. In the context of rare metabolic disorders, non-genetic factors- including dietary interventions- may represent disease modifiers that influence the variable phenotypic outcomes of inherited pathogenic variants.

## Methods
### Antibodies
The following antibodies were used: mouse anti-MT-CO1 (Abcam, ab14705); mouse anti-tubulin (Sigma, T6074); mouse anti-NDUFS3 (Abcam, ab14711); mouse anti- α-tubulin (Sigma, T8203); mouse anti-GFP (Roche, 11814460001); mouse anti-actin (Sigma, A5316); anti-TOM20 (Proteintech, 11802-1-AP), goat anti-rabbit Alexa Fluor™ 568 (ThermoFisher scientific, A11011), anti-rabbit (LI-COR Biosciences, IRDye® 800CW), and anti-mouse (LI-COR Biosciences, IRDye® 680RD).

### ATP measurements
Control and *wah-1* mutants expressing *wuIs305[myo-3p::Queen-2m]* were imaged with a 40x oil immersion objective using Airyscan Zeiss LSM900 confocal microscopes. The measurement of cytosolic ATP level was obtained by a ratiometric analysis of Queen-2m excitation signals at 405 and 488 nm. As a positive control, animals were treated with 10% NaN$_3$ for 5 min.

### Caenorhabditis elegans strains and maintenance
Nematodes were grown on nematode growth media (NGM) plates previously seeded on a bacterial lawn of OP50 *E. coli* and

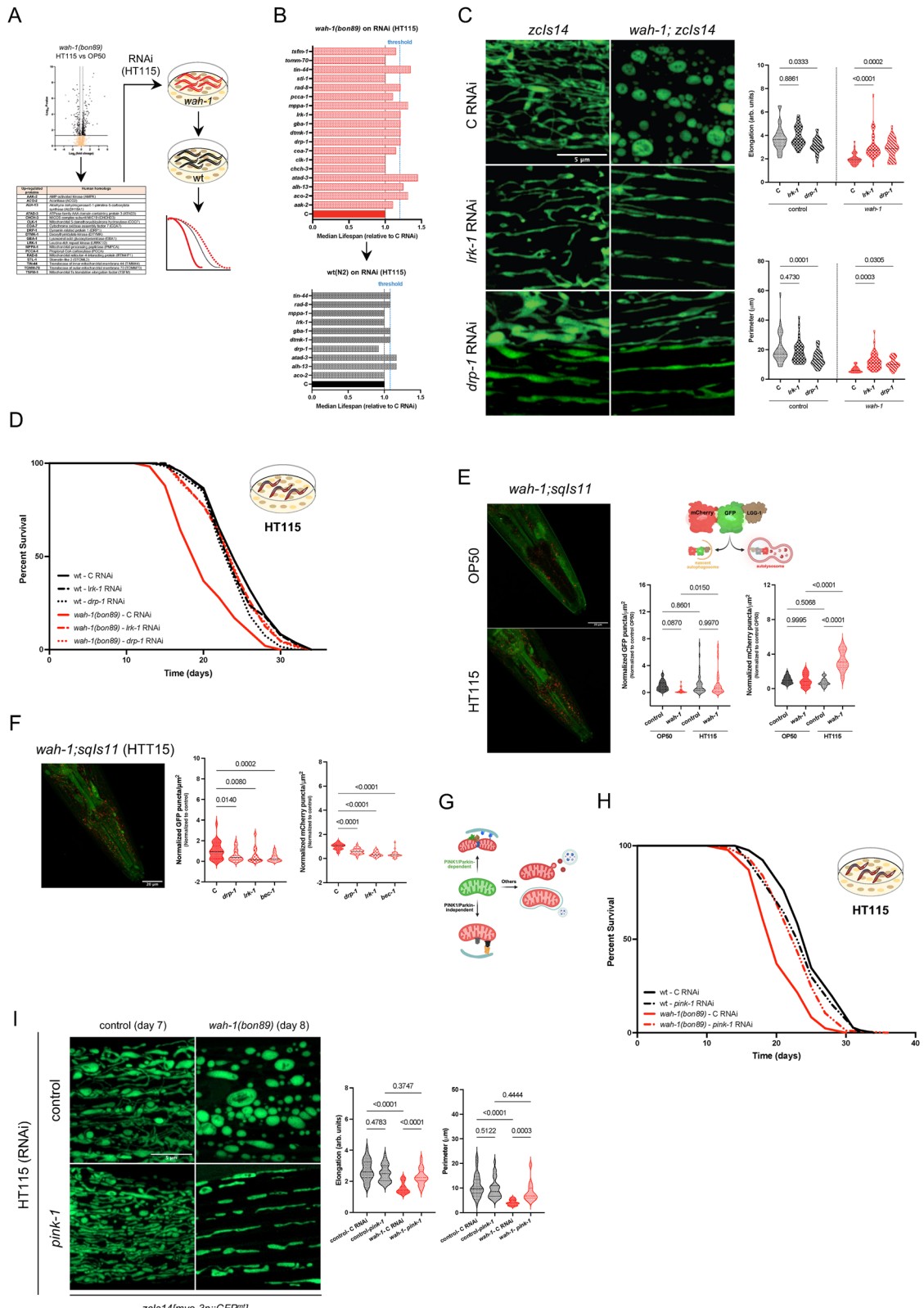

maintained in a temperature-controlled incubator at 20 °C. The following strains were used in this study: wild type N2 (Bristol), BAN1 *daf-2(e1370)*, BAN456 *wah-1(bon89)III*, BAN463 *wah-1(bon89)III;zcIs14 [myo-3p::GFP(mt)]*, BAN482 *daf-2(e1370);wah-1(bon89)*, BAN524 *idrIs1 [dhs-3p::dhs-3::gfp]*, BAN526 *wah-1(bon89)III*, BAN532 *bcIs39[lim-7p::ced -1::GFP+lin-15(+)]*, BAN528 *wah-1(bon89);bcIs39[lim-7p::ced-1::GFP+lin -15(+)]*, BAN592 *wah-1(bon89)III;idrIs1[dhs-3p::dhs-3::gfp]*, BAN744 *wah*

*-1(bon89);drp-1(tm1108)*, BAN774 *wah-1(bon89);sqIs11[lgg-1p::mCherry::GFP::lgg-1+rol-6]*, BAN797 *lrk-1(bon150)*, BAN798 *lrk-1(bon151)*, BAN801 *wah-1(bon89);drp-1(or1941[drp-1::GFP])*, BAN811 *lrk-1(bon150);wah-1(bon89)*, BAN812 *lrk-1(bon151);wah-1(bon89)*, BAN 861 *wah-1(bon89);zcIs13[hsp-6p::GFP]*, BAN862 *wah-1(bon89);zcIs9[hsp-60p::GFP +lin-15(+)]*, BAN 863 *wah-1(bon89);zcIs39[dve-1p::dve-1::GFP]*, BAN864 *wah-1(bon89);dvIs19[pAF15(gst-4p::GFP::NLS)]*, BAN880 *wah-1(bon89);*

**Fig. 4 | LRK-1 and DRP-1 regulate mitochondrial morphology and autophagy in hypomorphic *wah-1* mutants. A** Schematic representation of the HT115 RNAi screening against upregulated metabolic enzymes and mitochondrial-associated proteins in *wah-1(bon89)*. **B** Median lifespan (relative to control) of *wah-1(bon89)* animals grown on the indicated RNAi. Out of the 18 clones, 10 RNAi (red bars) increase *wah-1* survival by at least 20% ($n = 1$). As a counter-screening, the ten positive hits were tested in wt(N2) nematodes ($n = 1$). **C** Representative confocal images of mitochondrial network in control and *wah-1* mutants grown on control, *lrk-1* and *drp-1* RNAi (HT115 bacteria). Mitochondrial morphology is measured by elongation score and perimeter ($n = 30$-$40$ animals/condition from 3 biological replicates, one-way ANOVA; Dunnett' s multiple comparisons test).

**D** Representative survival curves of wt (N2) and *wah-1(bon89)* animals grown on control, *lrk-1* and *drp-1* HT115 RNAi bacteria. **E** Confocal images of *C. elegans* heads expressing *sqIs11[lgg-1p::mCherry::GFP::lgg-1+ rol-6]*. Scheme depicts the fluorescence quenching of mCherry::GFP::LGG-1 fusion proteins within an autolysosome. Created in BioRender. Bano, D. (2026) https://BioRender.com/xbjr3z5. The plot

shows the quantification of mCherry-positive (right violin graph) and GFP-positive puncta (left violin graph) ($n = 33$–$36$ animals from 3 independent experiments, one-way ANOVA, Šídák's multiple comparisons test). **F** Confocal image of *wah-1* mutant expressing *sqIs11[[lgg-1p::mCherry::GFP::lgg-1+ rol-6]*. Analysis of GFP-positive (left violin graph) and mCherry-positive (right violin graph) puncta in *wah-1(bon89)* animals grown on control, *drp-1, lrk-1* and *bec-1* (as an inhibitor of autophagy) RNAi (HT115). Statistics of GFP/$\mu m^2$ and mCherry-positive puncta are reported ($n = 30$-$31$ animals/condition from 3 biological replicates, one-way ANOVA, Dunnett's multiple comparisons). **G** Schematic representation of mitophagy pathways. Created in BioRender. Bano, D. (2026) https://BioRender.com/xbjr3z5. **H** Representative lifespan assay of wt and *wah-1(bon89)* animals grown on control and *pink-1* RNAi (HT115). (**I**) Confocal images of control and *wah-1* mutants expressing *zcIs14[myo-3p::GFP(mt)]*. Animals were grown on control and *pink-1* RNAi (HT115) since hatching. Quantification of mitochondrial elongation and perimeter is shown on the right ($n = 27$-$31$ animals/condition from 3 biological replicates; one-way ANOVA, Šídák's multiple comparisons test).

---

*atfs-1(tm4525)*, BAN962 *wah-1(bon89);wuIs30S[myo-3p::Queen-2m]*, BAN966 *unc-119(ed3);bonEx174[drp-1p::wrmScarlet::DRP-1::drp-1u+Cbr-unc-119(+)]*, BAN967 *atfs-1(tm4525)*, CL2166 *dvIs19[pAF15(gst-4::GFP::NLS)]*, CU6372 *drp-1(tm1108)*, EU2917 *drp-1(or1941[drp-1::GFP])*, GA2001 *wuIs30S[myo-3p::Queen-2m]*, MAH215 *sqIs11[lgg-1p::mCherry::GFP::lgg-1+rol-6]*, SJ4100 *zcIs13[hsp-6p::GFP]*, SJ4058 *zcIs9[hsp-60p::GFP+lin-15(+)]*, SJ4103 *zcIs14[myo-3p::GFP(mt)]*, SJ4197 *zcIs39[dve-1p::dve-1::GFP]*. Some strains were provided by the CGC, which is funded by the NIH Office of Research Infrastructure Programs (P40 OD010440). Newly developed *C. elegans* strains are fully available from the corresponding author upon request.

## Cell culture
Wild-type (Abcam, ab255449) and AIF KO HEK293T cells (Abcam, ab266347) were cultured in DMEM (Gibco, 10567014) containing 10% FBS. For the cell viability assay, wild-type and AIF KO HEK293T cells were seeded in a PDL-coated 24-well plate with DMEM containing 10% FBS. After 48 h, cells were washed and the medium was replaced with glucose-free DMEM (Gibco, A14430-01) supplemented with 1 g/L galactose and 2 mM glutamine (ThermoFisher, 35050038), with or without 120 μM oleic acid (Sigma, O3008). The plate was then placed in the IncuCyte to monitor cell growth and viability over 72 h. For the assessment of mitochondrial morphology, cells were seeded onto PDL-coated glass coverslips. After 72 h, the medium was replaced with glucose-free DMEM supplemented with 1 g/L galactose and 2 mM glutamine, with or without 120 μM oleic acid for a duration of three hours. Subsequently, the cells were fixed with 4% PFA, mitochondria were stained with TOM20 antibody (dilution 1:100), and imaged using a 63x oil immersion objective at the LSM900 Airyscan joint-deconvolution microscope. Images were further visualized and analyzed in ImageJ2 software.

## Confocal imaging of C. elegans
Mitochondrial morphology was assessed in GFP overexpressing animals using an Airyscan Zeiss LSM800 or LSM900 confocal microscopes equipped with a 63x oil immersion objective. In vivo quantification of lipid droplets was carried out in adult animals carrying the *idrIs1[dhs-3p::dhs-3::gfp]* transgene. CED-1::GFP positive engulfed cells were assessed in *bcIs39[lim-7p::ced-1::GFP+lin-15(+)]* expressing animals. Autophagy was quantified by using *sqIs11[lgg-1p::mCherry::GFP::lgg-1 + rol-6]*. Images were visualized and analyzed using ImageJ2.

## Lifespan assays
Young adult gravid animals were bleached with a hypochlorite solution, and eggs were transferred to NGM plates previously seeded with bacteria. Animals were grown at 20 °C and transferred to freshly

seeded bacteria every other day until all animals were dead. Control, *aak-2, aco-2, alh-13, atad-3, chch-3, clk-1, coa-7, dtmk-1, gba-1, mppa-1, pcca-1, pink-1 rad-8, stl-1, tin-44, tomm-70, tsfm-1, drp-1, lrk-1, skn-1, wah-1* RNAi clones were in *E. coli* HT115 (Ahringer library, Source Bioscience LifeSciences) or OP50(xu363) and were prepared as previously described[34]. For the metabolite supplementation experiments, 10 g/L glucose, 200 nM vitamin B12, and 500 μM oleic acid (along with 0.1% NP40) were added on NGM plates.

## Lipidomics
OP50, HT115, control, and *ΔdgkA* BW25113 *E. coli* were grown until $OD_{600} = 0.3$, followed by plating on NGM plates. After the plates were dried, they were incubated for 48 h at 20 °C before being collected for lipidomic analysis. For lipidomic analysis of wt and *wah-1(bon89)*, young adult gravid animals were bleached, and eggs were transferred to NGM plates that were previously seeded with OP50, HT115, control,l and *ΔdgkA* BW25113 *E. coli*. Then, adult nematodes were collected and processed for lipidomic analysis. Raw data are available in the Supplementary tables. Samples were prepared according to MetaSysX standard procedure and a modified protocol[76]. Between 50 and 100 μl of E. coli or nematodes were used for lipid extraction. The samples were then analyzed with a Waters ACQUITY Reversed Phase Ultra Performance Liquid Chromatography (RP-UPLC) coupled to a Thermo-Fisher Exactive mass spectrometer. Chromatograms were acquired using dd-MS2 Top 3 mode (Data Dependent tandem mass spectrometry) with the following settings: Full Scan MS mode (mass range 300-1500) and NCE 25 (Normalized Collision Energy). Acyl composition of DAGs and TAGs was determined by detecting precursor ion [M + H]+ and neutral losses [Acyl+NH4]+ in positive ion mode with the further combinatorial calculation of the acyl composition. Acyl-composition of phosphoglycerolipids was concluded from the mass-to-charge ratio of [Acyl-H]-fragments of the corresponding precursors in negative ion mode. Acyl composition of sphingolipids was established from the fragmentation pattern of [M + H]+ precursor ion in positive ionization mode. The LC-MS data were obtained with the software PeakShaper (MetaSysX GmbH). After extraction from the chromatograms, the data is processed, aligned, and filtered for redundant peaks using in-house software. LC-MS lipophilic peaks were assigned according to the in-house MetaSysX database of chemical compounds. Pure compounds were used as references. The MetaSysX developed-R-based algorithm was used for lipid annotation. Lipid content was quantified using liquid chromatography-coupled to mass spectrometry (LC-MS), which enables comparison of relative abundances of the same lipid species across samples. Differences in lipid composition between the two genotypes were reflected in the intensities of lipid ions. Relative quantitation by mass spectrometry was sufficient to identify significantly altered lipids between samples. For statistical analysis, normalized

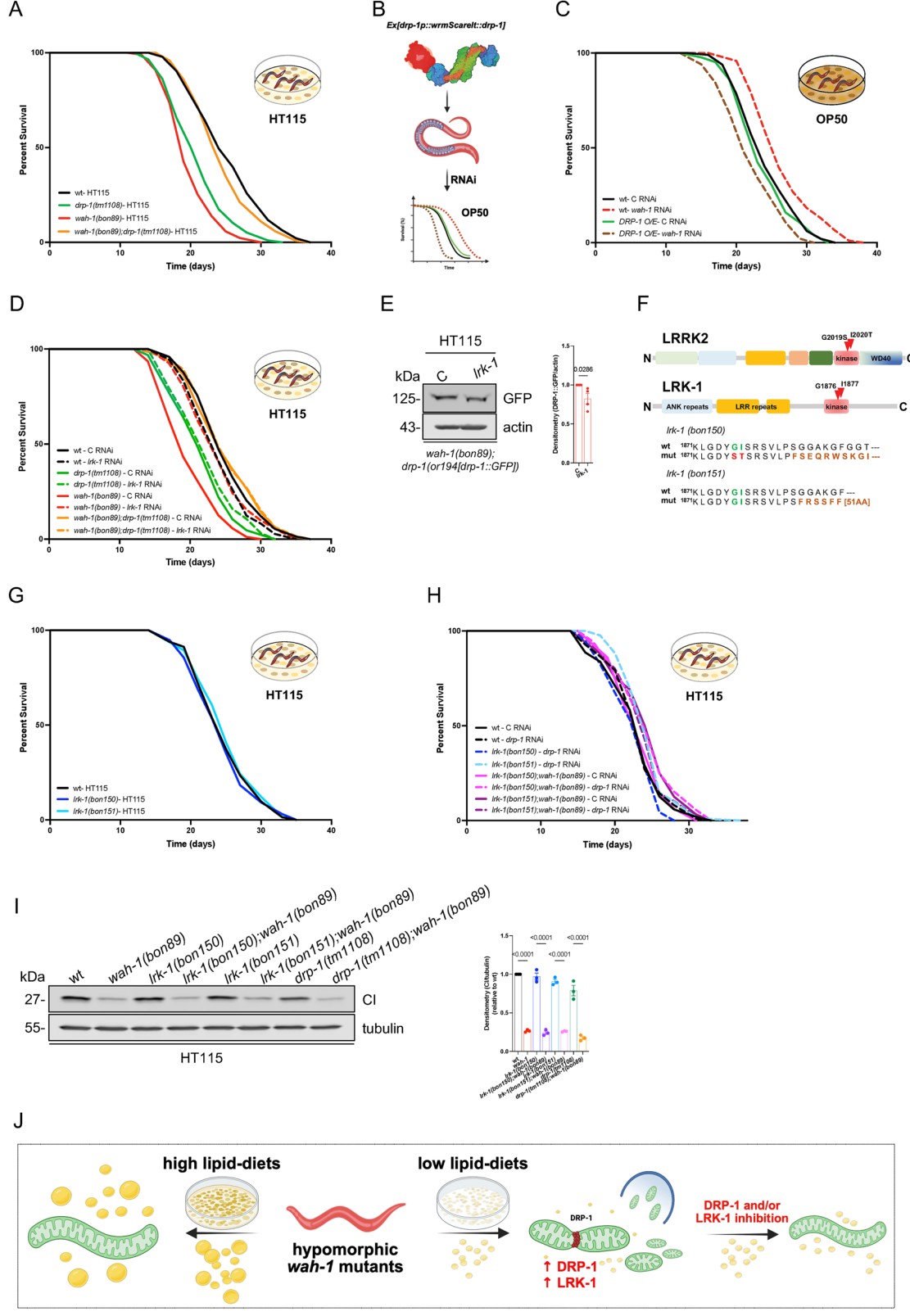

intensities were log2-transformed, and values were median-centered prior to Principal Component Analysis (PCA). The statistical tests were performed using a two-tailed t-test assuming unequal variance, and the resulting *p*-values were further adjusted using the Benjamini-Hochberg (BH) method. In all comparative analyses, molecules that were below the detection limit in one sample, but detectable in the other, were indicated as "not detected (n.d.)".

## Oxygen consumption rate measurements

OCR was measured using a Seahorse XFe24 Analyzer (Agilent). Nematodes were grown at 20 °C on OP50 plates until they reached the first day of adulthood. Then, adult animals were transferred to heat-killed OP50 plates for 3 h to purge their guts of any live bacteria. For each measurement, 50 animals were transferred to a well of an XFe24 plate containing 500 μL M9 buffer. Mitochondrial respiratory capacity

**Fig. 5 | LRK-1 and DRP-1 act on the same pathway to regulate *wah-1* lifespan.**
**A** Lifespan assays of wt (N2), *drp-1(tm1108)*, *wah-1(bon89)* and *wah-1(bon89);drp-1(tm1108)* animals on HT115. **B** Schematic representation of the wormScarlet-tagged DRP-1. The expression in *C. elegans* postmitotic cells is mediated by an extra-chromosomal array. To maintain the extrachromosomal array, an *unc-119(+)* rescue cassette was used in an *unc-119(ed3)* background. Since *wah*-1 and *unc-119* loci are relatively close, lifespan assays were performed with control and *wah-1* RNAi bacteria. Created in BioRender. Bano, D. (2026) https://BioRender.com/xbjr3z5.
**C** Lifespan assays of wt and DRP-1 overexpressing (O/E) animals on control and *wah-1* RNAi bacteria (OP50). **D** Representative survival curves of wt, *drp-1(tm1108)*, *wah-1(bon89)* and *wah-1(bon89);drp-1(tm1108)* animals on control and *lrk-1* RNAi bacteria (HT115). **E** Representative immunoblots of samples from *wah-1* mutants expressing *drp-1(or1941[drp-1::GFP])* grown on control and *lrk-1* RNAi bacteria (HT115). Actin was used as a loading control. Densitometry is reported on the right (*n* = 4, two-tailed t-test, Mann-Whitney test). **F** Scheme reports the main domains of

human LRRK2 and *C. elegans* LRK-1. Red arrowheads indicate two pathogenic mutations in LRRK2 (G2019S and I2020T) and the corresponding residues in LRK-1 (G1876 and I1877). The two newly generated *lrk-1* alleles and the predicted amino acid sequences (based on DNA sequencing) are reported at the bottom.
**G** Representative survival assay of wt, *lrk-1(bon150)* and *lrk-1(bon151)* animals on HTT15 bacteria. **H** Representative survival curves of wt (N2), *lrk-1(bon150)*, *lrk-1(bon151)*, *lrk-1(bon150);wah-1(bon89)* and *lrk-1(bon151);wah-1(bon89)* animals grown on control and *drp-1* RNAi bacteria (HT115). **I** Representative immunoblots of NUO-2/NDUFS3 (CI) and tubulin (as a loading control) of samples are from animals grown on HT115 bacteria. Densitometry is reported on the right (*n* = 3, Mean +/− SEM, one-way ANOVA, Šídák's multiple comparisons test). **J** Schematic summary of our findings. Low-lipid diets promote the expression of LRK-1 and DRP-1, which negatively impact mitochondrial network maintenance and survival. Created in BioRender. Bano, D. (2026) https://BioRender.com/xbjr3z5.

was measured as a result of a sequential exposure to 20 μM FCCP and 20 mM sodium azide as described in some of our prior studies[34,63–65].

## SDS-PAGE and western blotting

Animals were washed and lysed in RIPA buffer (Abcam, ab206996) supplemented with protease inhibitor (Roche, 11836153001) and phosphatase inhibitor (Roche, 04906837001). An equal amount of protein samples was mixed with 4x loading buffer (250 mM Tris-HCl, pH 6.8, 8% SDS, 40% glycerol, 20% 2-Mercaptoethanol, 0.01% bromo-phenol blue) and boiled at 95 °C for 5 min. Following, the samples were loaded on polyacrylamide gels and subsequently transferred onto nitrocellulose membranes. The membrane was first incubated with blocking solution (5% dried skim milk in TBST: 50 mM Tris-HCl, 150 mM NaCl, 0.1% Tween20) and subsequently with primary antibody solution (dilution 1:1000) overnight at 4 °C. The membrane was then incubated with fluorescence-tagged secondary antibody (dilution 1:10000) and imaged with Odyssey Infrared Imaging System (Li-Cor Biosciences) and quantified using ImageStudioLite (Li-Cor Biosciences).

## Quantitative real-time PCR

RNA extraction from adult animals was carried out using the RNeasy RNA extraction kit (Qiagen), and cDNAs were prepared using qScript cDNA Supermix (Quanta Biosciences). Further, quantitative RT-PCR was performed in a Step One Plus Real Time PCR System (Applied Biosystems) using Fast SYBR Green Master Mix (Applied Biosystems). *β-actin* was used for normalization. Fold changes of mRNA were analyzed using the comparative ΔΔCt method. The following oligonucleotides were used in this study: *β-actin* 5′-tgtgatgccagatcttctccat-3′ and 5′-gagcacggtatcgtcaccaa-3′; *wah-1* 5′- gctgatgctgtcgaggaga-3′ and 5′-tggtggtgttctcttctgtaga-3′; *mt-nd1* 5′-gtttatgctgctgtaagcgtg-3′ and 5′-ctgttaaagcaagtggacgag-3′; *cox1/mt-co1* 5′-tgggttgacaggtgttgtattatct-3′ and 5′-gtgtaacacccgtgaaaatcc-3′.

## Sample preparation, LC-MS/MS measurements, and SureQuant proteomics

Approximately 800 wt (N2) and *wah-1(bon89)* adult nematodes were collected at days 4 and 5 after hatching, respectively. Animals were washed twice with water, and the pellets were kept at -80 °C. Samples were lysed in 200 μL Lysis buffer (50 mM HEPES (pH 7.4), 150 mM NaCl, 1 mM EDTA, 1.5 % SDS, 1 mM DTT; supplemented with 1× protease and phosphatase inhibitor cocktail (ThermoScientific)). Lysis was aided by repeated cycles of sonication in a water bath (6 cycles of 1 min sonication (35 kHz) intermitted by 2 min incubation on ice). At least 25 μg of *C. elegans* protein lysates were reduced and alkylated prior to processing by a modified filter-aided sample preparation (FASP) protocol as previously described[77]. Samples were digested with Trypsin (1:20; in 50 mM ammonium bicarbonate) directly on the filters, precipitated using an equal volume of 2 M KCl for depletion of residual

detergents, then cleaned and desalted on C18 stage tips. Peptides for TMT-based MS analysis were re-suspended in 20 μL of 50 mM HEPES (pH 8.5). Then, wt (N2) and *wah-1(bon89) C. elegans* peptides were labeled with 25 μL of diluted (14.75 mM) 128 N, 128 C, 129 N and 129 C (clone 1), 130 N (clone7) TMT 10plex, respectively, for 1 h at RT. TMT signal was quenched by the addition of 2 μL of 5% hydroxylamine to the reaction, vortexing for 20 s and incubating for 15 min at 25 °C with shaking (1000 rpm). TMT-labeled peptides were acidified with 45% (vol/vol) of 10% FA in 10% ACN, prior to combining samples at equal amounts for drying in a concentrator. Dried peptides were re-suspended in 300 μL of 0.1% TFA for subsequent high pH reverse phase fractionation (Perce kit). 6 peptide fractions (10, 15, 20, 25, 50, and 80%) ACN were collected, concentrated, and re-suspended in 20 μL 5% FA for LC-MS analysis. MS runs were performed in triplicate. Peptides for label-free MS analysis were re-suspended in 20 μL 1% FA for LC-MS/MS analysis.

Tryptic peptides were analyzed on a Dionex Ultimate 3000 RSLC nanosystem coupled to an Orbitrap Exploris 480 MS (Thermo-Scientific). They were injected at starting conditions of 95% eluent A (0.1% FA in water) and 5% eluent B (0.1% FA in 80% ACN). Peptides were loaded onto a trap column cartridge (Acclaim PepMap C18 100 Å, 5 mm×300 μm i.d., #160454, Thermo Scientific) and separated by reversed-phase chromatography on an Acclaim PepMap C18, 100 Å, 75 μm X 50 cm (both columns from Thermo Scientific) using a 90 min linear increasing gradient from 8% to 25% of eluent B followed by a 30 min linear increase to 50% eluent B. The mass spectrometer was operated in data-dependent and positive ion mode with MS1 spectra recorded at a resolution of 120k with an automatic gain control (AGC) target value of 300% ($3{\times}10^6$) ions, maxIT set to Auto, and an intensity threshold of $1 \times 10^4$, using a mass scan range of 350−1550. Precursor ions for MS/MS were selected using a top speed method with a cycle time of 2 ms and normalized collision energy (NCE) of 36% (High-energy Collision Dissociation (HCD)), to activate both the reporter and parent ions for fragmentation. MS2 spectra were acquired at 45k resolution using an AGC target value of 200% ($2 \times 10^5$), and maxIT set to 86 ms. Dynamic exclusion was enabled and set at 45 s. Isolation width was set at 0.7 m/,z and the fixed first mass was set to 110 m/z to ensure reporter ions were detected. Peptide match was set to off, and isotope exclusion was on. Charge-state exclusion rejected ions that had unassigned charge states, were singly charged, or had a charge state above 5. Full MS data were acquired in the profile mode with fragment spectra recorded in the centroid mode.

TMT-based MS raw data files were processed with Proteome Discoverer™ software (v2.4.0.305, Thermo Scientific) using SEQUEST® HT search engine against the Swiss-Prot® *Caenorhabditis elegans database* (v2020-08-24). Peptides were identified by specifying trypsin as the protease, with up to 2 missed cleavage sites allowed. Precursor mass tolerance was set to 10 ppm, and fragment mass tolerance to

0.02 Da MS2. Static modifications were set as carbamidomethylated cysteine and TMT6plex (229.163 Da; N-terminal, K), while dynamic modifications included methionine, oxidation, and N-terminal protein acetylation, for all searches. Resulting peptide hits were filtered for a maximum 1% FDR using the Percolator algorithm. The TMT10plex quantification method within Proteome Discoverer software was used to calculate the reporter ratios with mass tolerance ±10 ppm, and applying isotopic correction factors. Only peptide spectra containing all reporter ions were designated as "quantifiable spectra". Label-free MS raw data files were processed with Proteome Discoverer™ software (v3.0 SP1, Thermo Scientific) using SEQUEST® HT search engine against the Swiss-Prot® *Caenorhabditis elegans database* (v2023-06-28). Peptides were identified by specifying trypsin as the protease, with up to 2 missed cleavage sites allowed and restricting peptide length between 7 and 30 amino acids. Precursor mass tolerance was set to 10 ppm, and fragment mass tolerance to 0.02 Da MS2. Static modifications were set as carbamidomethylated cysteine, while dynamic modifications included methionine and N-terminal loss of methionine, for all searches. Peptide and protein FDR were set to 1% by the peptide and protein validator nodes in the Consensus workflow. Default settings of individual nodes were used if not otherwise specified. In the Spectrum Selector node, the Lowest Charge State = 2 and Highest Charge State = 6 were used. The INFERYS rescoring node was set to automatic mode, and the resulting peptide hits were filtered for a maximum 1% FDR using the Percolator algorithm in the Processing workflow. A second-stage search was activated to identify semi-tryptic peptides. Both unique and razor peptides were selected for protein quantification. Proteins identified by site, reverse, or potential contaminants were filtered out prior to analysis.

Targeted MS analysis was performed using synthetic PEPotec isotope-labeled -C-terminal lysine (K) or arginine (R) crude peptides (Thermo Scientific). An equimolar amount of each heavy peptide was mixed together at a final concentration of 1 pmol/µl in 0.1% FA to generate a pool of wah-1 SIL peptides for subsequent nanoLC-MS/MS analysis to determine their intensities. SureQuant analysis was performed as previously described[78]. Briefly, data acquisition was performed using a modified SureQuant template with 3 branches for the +2 (R, K), +3 (K) charge states of SIL lysine and arginine residues. Peak area ratios of endogenous light peptides and corresponding heavy IS peptides for the 6 selected product ions were exported from Skyline software v21.1.0.278[79], and peptides were filtered according to the following criteria: First, only IS peptides with an AUC > 0 for $n \geq 5$ product ions were considered. Second, peak area values of the 3 highest intensity product ions from both the light/heavy peptides were summed, and their light/heavy ratios were used to quantify peptide signals across samples. Quantitation was based on 3 selected product ions to balance specificity with the ability to retain lowly abundant targets. Data have been deposited into a publicly available repository[80] (see Data Availability) and are also included as Supplementary table proteomics STP1-3.

Statistical analysis was performed in *Proteome Discoverer* (v2.4.0.305) for both TMT and LFQ datasets. Data were analyzed from four biological replicates. Protein abundances were $\log_2$-transformed prior to testing. Since only two biological groups were compared, differential protein expression was evaluated using a two-sided Student's t-test. Multiple hypothesis testing was controlled using the Benjamini–Hochberg false discovery rate (FDR). Proteins were considered significantly changed at $p < 0.05$ and $|\log_2$ fold-change $| \geq 0.33$ thresholds. Targeted MS (SureQuant) data were processed in Skyline, light/heavy ratios derived from peptide peak areas were $\log_2$-transformed and compared between groups using a two-sided t-test. No FDR correction was applied to the targeted MS data since very few peptides were analyzed. SQ analysis was done using three biological replicates.

## Statistics and reproducibility

No statistical method was used to predetermine sample size. No data were excluded from the analyses. The Investigators were not blinded to allocation during experiments and outcome assessment. For statistical analysis, GraphPad Prism Software was used. Student's t-test was used to compare between two groups, and one-way ANOVA (with appropriate post hoc correction) was employed for three or more groups. The number of biological replicates is indicated in the figure legends.

## Data availability

All numerical source data and uncropped images of western blots are published alongside the paper in a Source Data file. The LC-MS/MS data generated in this study have been deposited to publicly available repositories: accession number PXD046764 for ProteomeXchange and JPST002369 for jPOST. Source data are provided with this paper.

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

## Acknowledgements

We would like to thank Ms. Christiane Bartling-Kirsch (DZNE) and Ms. Lara Susan Weichelt (DZNE) for their technical support. Furthermore, we thank Ms. Magdalena Musialak-Lange (metaSysX GmbH) for the lipidomic analysis. This research was supported by the DZNE institutional budget. DB is a member of: the Deutsche Forschungsgemeinschaft (DFG, German Research Foundation) under Germany's Excellence Strategy – EXC2151 – 390873048, Excellence Cluster ImmunoSensation[2]; the ETERNITY project consortium, funded by the European Union through Horizon Europe Marie Skłodowska-Curie Actions Doctoral Networks (MSCA-DN) under the grant number 101072759. IMM is supported by the Deutsche Forschungsgemeinschaft (DFG, German Research Foundation) – SPP2453 project number 541647076 (DB: grant recipient). In accordance with the journal's guidelines, we declare that AI chatbots were used to refine sentence structure and English grammar.

## Author contributions

All authors contributed to the study and met the authorship criteria. All authors approved the final manuscript. M.M. Investigation, Formal analysis, Visualization. Enzo Scifo: Investigation, Formal analysis. R.E.C.: Investigation, Formal analysis. L.W.: Investigation, Formal analysis. T.N.A.: Investigation, Formal analysis. J.J.: Investigation, Formal analysis. I.-M.M.: Investigation. V.Z.-D.: Investigation, Formal analysis. J.R.: Reagents. B.J.: Reagents. C.E.H.: Reagents. S.K.: Investigation. L.S.: Investigation. P.N.: Funding acquisition. D.E.: Funding acquisition, Resources, Formal analysis. D.B.: Conceptualization, Formal analysis, Writing-Original Draft, Visualization, Supervision, Project administration, Funding acquisition.

## Funding

## Competing interests

The authors declare no competing interests.
