## [Transparent Peer Review file · Nature Communications]

Dietary lipid content modifies wah-1/AIFM1-associated phenotypes via LRK-1 and DRP-1 expression in *C. elegans*

Corresponding Author: Dr Daniele Bano

Version 0:

Reviewer comments:

Reviewer #1

(Remarks to the Author)

This manuscript investigates the impact of lipid availability on wah-1/AIFM1 mutant *C. elegans*, which inhibits the expression of mitochondrial Complex I subunits. Through a comprehensive experimental approach, the authors propose a novel mechanism involving LRK-1 and DRP-1 that modulates mitochondrial dynamics and longevity in response to dietary lipid intake. The effort invested in creating clear and well-structured figures is commendable. However, significant scientific and methodological issues undermine the validity and broader applicability of the findings to other species.

Major Concerns

- (1) Despite the relevance of mitochondrial dysfunction to human diseases, the manuscript lacks a robust justification for its relevance to human mitochondrial disorders. Minimal evidence is provided connecting *C. elegans* lipid metabolism and mitochondrial dynamics to human physiology, weakening the clinical significance of the findings. Including supporting evidence from mammalian models would greatly enhance the translational impact.
- (2) The exclusive reliance on *C. elegans* limits the broader applicability of the results. Mitochondrial dynamics and lipid metabolism in this invertebrate model may not fully represent those in mammals, particularly in tissues such as the brain and muscle, where mitochondrial dysfunction is most critical. If vertebrate models are currently unavailable, mammalian cell lines could be utilized to demonstrate that the wah-1/AIFM1 mechanism is conserved across species in response to lipid availability.
- (3) The study does not sufficiently validate that the effects of the hypomorphic wah-1 mutation on mitochondrial biogenesis are specific and not due to off-target or pleiotropic effects. Furthermore, the authors assume different lipid contents in the *E. coli* strains (OP50 and HT115) without biochemical quantification. Complementation assays or genetic rescue experiments are necessary for a detailed characterization of the dietary components.
- (4) The lifespan assays, proteomics, and lipidomics results lack sufficient detail on statistical methodologies. Comprehensive descriptions of statistical analyses should be included to ensure robustness and reproducibility.
- (5) Some figures lack clear legends, and data points are inconsistently labeled, making interpretation challenging. Improving figure clarity with detailed legends and consistent labeling across datasets is essential for accurate data presentation.
- (6) The manuscript does not explore alternative pathways or feedback mechanisms that could influence mitochondrial behavior, such as oxidative stress pathways or nutrient-sensing networks. Including potential regulatory feedback loops and alternative pathways would broaden the readers' understanding and provide a more holistic view of mitochondrial dynamics.
- (7) The authors assert that diet-genotype interactions significantly modify disease phenotypes, but this conclusion is based solely on *C. elegans*. To ensure a safe and generalized conclusion, results should be interpreted within the *C. elegans* research scope.

Reviewer #2

(Remarks to the Author)

In this manuscript, Mondal and colleagues reported an interesting interaction between a disease-related genetic mutation and environmental nutritional factors. RNAi or hypomorphic mutation of wah-1 shortens animal lifespan when fed with HT115, but prolongs the lifespan when fed with OP50. Further phenotypic analyses revealed that wah-1 mutant animals exhibit compromised mitochondrial morphology and function on the HT115 diet, with mitochondrial Complex I primarily affected. By comparing the compositions of these two diets and utilizing other *E. coli* strains (dgkA), the authors concluded

that high lipid diets are beneficial for wah-1-mutated animals possibly through alleviating mitochondrial stress-induced energetic burden. Furthermore, LRK-1/ DRP-1-induced mitochondrial fission and autophagy (most likely mitophagy) are also involved in sensitizing wah-1-mutated animals to the HT115 low-lipid diet.

The writing is concise and clear. The research is also timely, as interactions between genetic mutations and environmental factors are receiving increasing attention in the field. More specifically, this study will inspire the fields of mitochondria and mitochondrial diseases, offering insights into how to maintain mitochondrial homeostasis under different mitochondrial stress contexts using distinct strategies.

Addressing the following points may help further solidify the conclusions with more convincing mechanistic details:

Major concerns:

1. Fig 1. Wah-1 mutant worms are longer lived on OP50. What's the mechanism for that?
Mitohormesis can be one explanation. The authors may check mitochondrial stress-related pathways in control and mutant worms on OP50 and HT115, either using RNA-seq, or more specifically, by crossing with specific mitoUPR or oxidative stress response reporters such as hsp-6/hsp-60/dve-1/gst-4, then block these pathways to see whether the longevity phenotype on OP50 can be abrogated.
2. As an extension to point 2, the authors may further check whether feeding wah-1 mutant animals with high lipid diet (dgkA) may rescue the expression of those mitochondrial stress reporters, Complex I levels, and OCR. Furthermore, as a connection between lipids and mito fission/mitophagy (Fig 3 and 4), whether dgkA feeding can also suppress DRP-1 expression in wah-1 mutant animals?
3. It seems that WAH-1 very specifically regulates complex I expression or activity in *C. elegans*. What is the mechanism about such high specificity? Previous reports show that AIFM1 promotes translocation of certain Complex I subunits into mitochondria (Salscheider, et al. 2022). The authors may further explain this in the text, or even design some experiments to confirm this to help readers understand Fig 1 better.
4. Is total amount of mitochondria reduced in wah-1 mutant worms on OP50 and HT115? More mitochondrial markers can be measured by western blot or mtDNA can be quantified.
5. Fig. 4. Mitochondrial fission and mitophagy genes seem to be required for the short lifespan phenotype of wah-1 mutant animals on HT115. The authors have confirmed that mitochondrial morphological changes can be reversed by knocking down *lrk-1* and *drp-1* (Fig 4). What about other parameters of mitochondria as described in earlier figures (such as OCR and Complex I expression), and lipid metabolism?
6. Fig. 4. It would be more convincing to test whether mitophagy, instead of autophagy, is also higher in wah-1 mutant animals on HT115 and declined upon *lrk-1* and *drp-1* RNAi treatment.
7. Fig 4E shows that DRP-1 expression is induced in wah-1 mutant animals particularly under the HT115 condition, which might be a consequence of increased mitochondrial stress. Therefore, the title "LRK-1 and DRP-1 sense lipid availability" seems not precise enough, because they are not direct sensors of lipids.
8. According to the model proposed by authors (line 305), lower TAG levels from HT115 diet may exacerbate the energy crisis due to WAH-1 deficiency. This can be tested by measuring ATP levels in control and mutant worms fed with OP50, HT115, or dgkA.

Minor points

9. Fig. 1J. Wah-1 mutant animals have reduced OCR. Do they have increased glycolysis to maintain their energy supply?
10. Fig. 2H. This result is rather distractive and dispensable to major conclusions. It may be moved to the supplementary figure. Moreover, the images are not convincing to support that wah-1 is mainly expressed in the intestine "compared to other tissues". It is possible that this gene is also highly expressed elsewhere, but the signals are just masked by the bulky intestine tissue. The authors may check published single cell RNAseq databases for this gene expression pattern to confirm this point.
11. The title is a bit confusing and shall be modified to reflect major conclusions. So far the mechanism with "LRK-1 and DRP-1" is not clear enough, but dietary lipids is a very convincing point.

Reviewer #3

(Remarks to the Author)

The authors provided several lines of evidence to suggest that LRK-1 and DRP-1 act in the same pathway to sense diet-derived lipids and modify wah-1/AIFM1-associated disease phenotypes. This finding is interesting and will have impact in the field. Several links are still required to complete this finding. For example, the manuscript did not present sufficient data to support the relationship between diet-derived lipids and LRK-1 and DRP-1. The data related to LRK-1 and DRP-1 should also be verified in wah-1(bon89) fed OP50. Furthermore, wah-1(bon89) nematodes fed OP50 and HT115 had less green fluorescence than WT in Figure 2, indicating lower lipid content in wah-1(bon89), but they exhibit different longevity changes compared to WT.

Version 1:

Reviewer comments:

Reviewer #1

(Remarks to the Author)

The authors convincingly addressed nearly all major concerns. The inclusion of mammalian cell line data, additional pathway analyses, and improved statistical clarity significantly strengthen the manuscript. In conclusion, although some limitations remain inherent to the *C. elegans* model, the revisions and responses are satisfactory.

Reviewer #2

(Remarks to the Author)

The authors have sufficiently addressed my concerns from the initial review.

Reviewer #3

(Remarks to the Author)

The revised manuscript included new results to respond the reviewer's comments, which makes the work to be suitable for publication.

Nature Communications manuscript NCOMMS-24-70146

Point-by-point response to the reviewers' comments.

We sincerely thank both reviewers for the time and effort they dedicated to evaluating our manuscript. We have carefully considered all their constructive feedback and performed additional experiments to address their comments. Below, we provide a detailed, point-by-point response to each of their suggestion and concern. Given the substantial amount of new data obtained during the revision, we had to significantly restructure the text, tables and representative figures (78 multi-panels in total). We hope that the revised manuscript meets the Editor's and reviewers' expectations and aligns with the standards required for publication in *Nature Communications*.

Reviewer #1 (Remarks to the Author):

This manuscript investigates the impact of lipid availability on *wah-1/AIFM1* mutant *C. elegans*, which inhibits the expression of mitochondrial Complex I subunits. Through a comprehensive experimental approach, the authors propose a novel mechanism involving LRK-1 and DRP-1 that modulates mitochondrial dynamics and longevity in response to dietary lipid intake. The effort invested in creating clear and well-structured figures is commendable.

We thank the reviewer for his/her remark.

However, significant scientific and methodological issues undermine the validity and broader applicability of the findings to other species.

Major Concerns

(1) Despite the relevance of mitochondrial dysfunction to human diseases, the manuscript lacks a robust justification for its relevance to human mitochondrial disorders. Minimal evidence is provided connecting *C. elegans* lipid metabolism and mitochondrial dynamics to human physiology, weakening the clinical significance of the findings. Including supporting evidence from mammalian models would greatly enhance the translational impact.

We would like to emphasize once more that inherited mutations in the human *AIFM1* gene are causally linked to primary mitochondrial diseases (Bano & Prehn, 2018; Reinhardt *et al*, 2020; Wischhof *et al*, 2022). This causal link has been confirmed clinically and experimentally, including in our transgenic mice (Wischhof *et al*, 2018). This information has been mentioned in several parts of our manuscript. However, if the reviewer suggests including additional details about pathogenic AIFM1 mutations and their link to mitochondrial disorders, we would be happy to expand the Introduction and Discussion sessions accordingly.

Over the past ten years, we and others have tried many times to generate partial loss of function mutations that could model *AIFM1*-related disorders. Two of these attempts, including an in-frame deletion of the Arg309 (corresponding to human Arg201), were previously reported (Troulinaki *et al*, 2018). These two mutations did not cause any obvious phenotype. Here, we have described an in-frame deletion in the *wah-1/AIFM1* locus that encodes a bona fide hypomorphic protein variant that causes mitochondrial Complex I dysfunction in *C. elegans*. As the reviewer may know, such a model was not available and researchers have so far relied on knockdown or knockout approaches. Our comprehensive unbiased analyses showed that the molecular signatures observed in nematodes were highly comparable to those observed in patients and transgenic mice, such as the Harlequin mutant and the *Aifm1* (*R200 del*) KI mice (Bertan *et al*, 2021; Wischhof *et al.*, 2018). Along with Complex I defects, hypomorphic *wah-1* mutants show defective expression of AIF/CHCHD4's substrates (e.g., LET-754/AK2, FAMH-136/FAM136A, TIN-13/TIMM13, DDP-1/TIMM8A) that undergo oxidative folding within the intermembrane space. Thus, while we acknowledge the limitations of our system, we have provided substantial experimental evidence that the newly generated hypomorphic *wah-1* mutation induces mitochondrial defects with molecular and phenotypic similarities to those observed in higher organisms.

(2) The exclusive reliance on *C. elegans* limits the broader applicability of the results. Mitochondrial dynamics and lipid metabolism in this invertebrate model may not fully represent those in mammals, particularly in tissues such as the brain and muscle, where mitochondrial dysfunction is most critical. If

vertebrate models are currently unavailable, mammalian cell lines could be utilized to demonstrate that the *wah-1/AIFM1* mechanism is conserved across species in response to lipid availability.

We are aware that every experimental model, including mice, cannot perfectly mimic human diseases. While our experimental data clearly demonstrate evolutionarily conserved molecular signatures (e.g., proteomic changes) and phenotypic changes (e.g., mitochondrial morphology), we have been very careful not to overinterpret the significance of our findings and/or their relevance to human disorders. If the reviewer feels that we have overstated any our conclusions, we would kindly appreciate his/her feedback and which specific points should be revised.

Having that said, the primary scope of our study was to identify molecular mechanisms that are causally linked to the phenotypic variability of a hypomorphic *AIFM1* mutation. We believe that this goal was achieved by leveraging *C. elegans* genetic tractability and by challenging the system with experimentally controlled interventions. In a consistent manner, we were able to show the mechanistic link between lipid availability, mitochondrial dynamics (DRP-1) and LRK-1 expression in a whole animal exposed to different diets.

Once more, we acknowledge the limitation of our study, since we could not translate our findings in our transgenic *Aifm1* (*R200 del*) KI mice due to discontinuation of our colony. However, motivated by the reviewer's comment and in agreement with the Editor, we have conducted some proof-of-principle experiments in cultured HEK293T cells. We were able to show that oleic acid supplementation could promote a more fused mitochondrial network in glucose-deprived, galactose-fed AIF KO HEK293 cells (Supplementary figure S2E). Remarkably, oleic acid supplementation could rescue the survival defects of AIF KO cells (Supplementary figure S2F), further suggesting that lipids can ameliorate phenotypes associated with AIF deficiency. We hope that the reviewer finds our evidence sufficient to support our conclusions and their potential relevance in mammalian cells.

(3) The study does not sufficiently validate that the effects of the hypomorphic *wah-1* mutation on mitochondrial biogenesis are specific and not due to off-target or pleiotropic effects. Furthermore, the authors assume different lipid contents in the *E. coli* strains (OP50 and HT115) without biochemical quantification. Complementation assays or genetic rescue experiments are necessary for a detailed characterization of the dietary components.

We would appreciate if the reviewer could kindly clarify his/her concern about "off-target or pleiotropic effects". The genetic tractability of *C. elegans* is a widely recognized strength of the model. All the phenotypes are linked to the mutation within the *wah-1/AIFM1* gene and have been partially recapitulated in animals exposed to *wah-1* RNAi. Based on our extensive proteomic analyses, molecular signatures clearly indicate an impairment of the AIF/CHCHD4 system and CI defects that resemble those observed in mammals.

If by "pleiotropic effects" the reviewer is referring to the possibility that WAH-1 deficiency promotes mitochondrial biogenesis by inducing downstream signaling pathway, we confirm that this is indeed the case. As a consequence of reduced AIF/CHCHD4 activity and subsequent mitochondrial OXPHOS inhibition, WAH-1 deficiency activates a mitochondria-to-nucleus stress response (Figure 2). Importantly, we show that SKN-1/Nrf plays a causal role in *wah-1* lifespan extension and mitochondrial network expansion in animals grown on OP50 bacteria.

With regard to the comments on lipids content in the *E. coli* strains, we initially employed two bacteria strains (OP50 and HT115) that have been extensively characterized at the metabolomic levels by other colleagues in the field (Brooks *et al*, 2009; Stuhr & Curran, 2020; Watts & Ristow, 2017). Instead of assuming a different lipid composition based on prior published works, we did provide an unequivocal snapshot of the differences between the two bacteria strains by carrying out state-of-the-art lipidomics that enable quantitative analyses of their distinct lipid contents. To further support our conclusions, we went on and added two additional bacteria strains, for which biochemical quantification of their metabolic profiles and effects on *C. elegans* have been provided in great details (Figure 3, Supplementary figures and Supplementary tables). Given that our primary aim was to examine the impact of lipid-rich diets on *C. elegans* survival, we believe that the data submitted for publication adequately support our conclusions.

We appreciate the reviewer's suggestion that an in-dept genetic rescue experiments or screening of an *E. coli* library (such as the Keio collection) could help to identify distinct metabolites and/or lipid species influencing the *wah-1* mutant *C. elegans*. Apart from being outside the primary aims of this study, we hope that the reviewer understands that undertaking such extensive analyses exceeds the scope of a standard peer-review request, as these approaches would require years of intense investigation given that our primary readout involves lifespan assays.

(4) The lifespan assays, proteomics, and lipidomics results lack sufficient detail on statistical methodologies. Comprehensive descriptions of statistical analyses should be included to ensure robustness and reproducibility.

As in our previous studies, we have fully adhered to international guidelines for data reporting. For the lifespan assays, conclusions are based on at least three biological replicates. In addition to representative graphs, we have included detailed Supplementary tables that report each experiment and describe the corresponding statistical analysis, both within the tables and in the Materials and Methods section.

For the proteomics and lipidomics analyses, we have provided representative graphs summarizing the main findings, along with comprehensive Supplementary tables containing all relevant experimental details for the benefit of the research community. In accordance with the journal's policy, our raw data have been deposited in a public repository.

In summary, we are confident that the reviewer's concerns have been thoroughly addressed.

(5) Some figures lack clear legends, and data points are inconsistently labeled, making interpretation challenging. Improving figure clarity with detailed legends and consistent labeling across datasets is essential for accurate data presentation.

Following the reviewer's comments, we have improved our panels and figure legends accordingly.

(6) The manuscript does not explore alternative pathways or feedback mechanisms that could influence mitochondrial behavior, such as oxidative stress pathways or nutrient-sensing networks. Including potential regulatory feedback loops and alternative pathways would broaden the readers' understanding and provide a more holistic view of mitochondrial dynamics.

To elucidate the mechanisms associated with WAH-1/AIF deficiency, our study has employed several unbiased approaches, including multi-omics and an RNAi screening. Most of our experimental validations were specifically designed to dissect novel signaling pathways. Following the reviewer's comments, we have now included additional data on oxidative stress and nutrient-sensing pathways. In line with our previously published data (Troulinaki K et al, 2018), we herein confirm that:

- Hypomorphic *wah-1* mutation does not influence insulin/IGF-1/DAF-2 signaling pathway (Supplementary figure S2B);
- Similar to other mitochondrial mutants, *wah-1* deficiency promotes "mitohormesis" and mitochondria-to-nucleus stress responses, resulting in the transcriptional activation of ATFS-1 (*hsp-6*, *hsp-60* and *dve-1* upregulation) and SKN-1 (*gst-4* upregulation and effects on lifespan) (Figure 2).
- Based on our proteomics (Figure 1E) and the GFP reporter lines (Figure 2), WAH-1 inhibition promotes GST-4 upregulation, which would suggest an increased glutathione synthesis to prevent oxidative stress.

We trust that these additional data fully address the reviewer's request.

(7) The authors assert that diet-genotype interactions significantly modify disease phenotypes, but this conclusion is based solely on *C. elegans*. To ensure a safe and generalized conclusion, results should be interpreted within the *C. elegans* research scope.

Once more, we fully agree with the reviewer that our conclusions are based primarily on *C. elegans* data and their clinical relevance must be conclusively determined in future studies. We have acknowledged the limitations of our study in the manuscript and moderated our conclusions accordingly. However, we are happy to further revise the text if the reviewer still believes that it would be appropriate.

Reviewer #2 (Remarks to the Author):

In this manuscript, Mondal and colleagues reported an interesting interaction between a disease-related genetic mutation and environmental nutritional factors. RNAi or hypomorphic mutation of *wah-1* shortens animal lifespan when fed with HT115, but prolongs the lifespan when fed with OP50. Further phenotypic analyses revealed that *wah-1* mutant animals exhibit compromised mitochondrial morphology and function on the HT115 diet, with mitochondrial Complex I primarily affected. By comparing the compositions of these two diets and utilizing other *E. coli* strains ($\Delta dgkA$), the authors concluded that high lipid diets are beneficial for *wah-1*-mutated animals possibly through alleviating mitochondrial stress-induced energetic burden. Furthermore, LRK-1/ DRP-1-induced mitochondrial fission and autophagy (most likely mitophagy) are also involved in sensitizing *wah-1*-mutated animals to the HT115 low-lipid diet.

The writing is concise and clear. The research is also timely, as interactions between genetic mutations and environmental factors are receiving increasing attention in the field. More specifically, this study will inspire the fields of mitochondria and mitochondrial diseases, offering insights into how to maintain mitochondrial homeostasis under different mitochondrial stress contexts using distinct strategies.

We would like to thank the reviewer for his/her kind words and greatly appreciate his/her supportive assessment of our study.

Addressing the following points may help further solidify the conclusions with more convincing mechanistic details:

Major concerns:

(8) Fig 1. *wah-1* mutant worms are longer lived on OP50. What's the mechanism for that? Mitohormesis can be one explanation. The authors may check mitochondrial stress-related pathways in control and mutant worms on OP50 and HT115, either using RNA-seq, or more specifically, by crossing with specific mitoUPR or oxidative stress response reporters such as *hsp-6/hsp-60/dve-1/gst-4*, then block these pathways to see whether the longevity phenotype on OP50 can be abrogated.

As mentioned in the original version, the “mitochondrial threshold effect” (or mitohormesis) was the starting point of our study and a continuation of our previous works (Troulinaki K et al, CDDiscovery 2018; Piazzesi A et al, Cell Reports 2016; Piazzesi A et al, Embo Reports 2022; Jackson J et al, Mol Metabolism 2022). Following the reviewer's comments, we have included a comprehensive analysis of the mitochondria-to-nucleus stress response. In this regard, we have provided data about the expression of *hsp-6*, *hsp-60*, *dve-1*, and *gst-4* in both control and *wah-1* mutants grown on OP50 and HT115. Additionally, we have tested the mechanistic impact of these pathways in *wah-1* mutant survival. Remarkably, we found that SKN-1 contributes to the lifespan extension of *wah-1* mutant animals in OP50 bacteria.

In the revised manuscript, we have now included a broader discussion of our findings in the context of “mitohormesis”.

(9) As an extension to point 2, the authors may further check whether feeding *wah-1* mutant animals with high lipid diet ($\Delta dgkA$) may rescue the expression of those mitochondrial stress reporters, Complex I levels, and OCR. Furthermore, as a connection between lipids and mito fission/mitophagy (Fig 3 and 4), whether $\Delta dgkA$ feeding can also suppress DRP-1 expression in *wah-1* mutant animals?

We thank the reviewer for his/her suggestions regarding the potential impact of a high lipid diet ($\Delta dgkA$) on mitochondrial stress reporters, Complex I, and DRP-1. All experiments have been performed and reported in our revision. Among other evidence, we have shown that:

- Compared to *E. coli* K-12 controls, $\Delta dgkA$ bacteria do not alter UPR^{mt} and Complex I defects, whereas they promote Skn-1/Nrf activity (Supplementary figure S3D-H).
- *E. coli* K-12 $\Delta dgkA$ inhibits DRP-1::GFP expression in *wah-1(bon89)* mutants (Figure 3J).

(10) It seems that WAH-1 very specifically regulates complex I expression or activity in *C. elegans*. What is the mechanism about such high specificity? Previous reports show that AIFM1 promotes translocation of certain Complex I subunits into mitochondria (Salscheider, et al. 2022). The authors may further explain this in the text, or even design some experiments to confirm this to help readers understand Fig 1 better.

We sincerely appreciate the reviewer's comments, which prompted us to re-examine our proteomics. Upon a closer look at our dataset, we herein confirm that WAH-1/AIF deficiency does indeed impair the expression of substrates that undergo oxidative folding within the intermembrane space. Among these mitochondrial precursors, we found LET-754/AK2 (as recently described in one of our recent papers: (Rothemann *et al*, 2025)), FAMH-136/FAM136A (Harhai *et al*, 2025; Wischhof *et al.*, 2018), (CX₃C)₂ motif-containing translocases (TIN-13/TIMM13 and DDP-1/TIMM8A), and (CX₉C)₂ motif-containing Complex I subunits (NDUB-7/NDUFB7 and NDUFS-8/NDUFS8) (Reinhardt *et al.*, 2020) (Figure 1E and Supplementary figure S1H).

(11) Is total amount of mitochondria reduced in *wah-1* mutant worms on OP50 and HT115? More mitochondrial markers can be measured by western blot or mtDNA can be quantified.

As the reviewer knows, only a small number of validated antibodies against *C. elegans* proteins can be used to assess changes in the mitochondrial proteome. To address the reviewer's request and substantiate our conclusions further, we have measured mtDNA in *wah-1* mutants grown on OP50 and HT115. Consistent with an abundant literature that describe mitochondrial biogenesis in animals carrying inherited mitochondrial lesions, *wah-1(bon89)* mutants have a higher mtDNA content compared to wt (N2) (Supplementary figure S2D). While the two diets may have a similar effect on mtDNA copy number, they have a different impact on mitochondrial morphology, as shown in various representative images. These data suggest that WAH-1/AIF deficiency is the trigger of mitochondrial biogenesis, independently of the diets. However, the maintenance of the mitochondrial network depends on diet-derived lipids, as discussed in various parts of our manuscript.

(12) Fig. 4. Mitochondrial fission and mitophagy genes seem to be required for the short lifespan phenotype of *wah-1* mutant animals on HT115. The authors have confirmed that mitochondrial morphological changes can be reversed by knocking down *lrk-1* and *drp-1* (Fig 4). What about other parameters of mitochondria as described in earlier figures (such as OCR and Complex I expression), and lipid metabolism?

Following the reviewers' comments, we assessed OCR, CI expression and lipid droplets in animals exposed to RNAi against *lrk-1* and *drp-1*. We found that:

- Inhibition of either *lrk-1* or *drp-1* did not rescue CI defects (Figure 5I). This is expected, as CI defects are due to aberrant WAH-1/AIF-CHCHD4/MIA40 activity.
- Consistently, neither *lrk-1* nor *drp-1* loss of function rescued mitochondrial respiration (see Supplementary panels for the reviewers, A).
- Downregulation of neither *lrk-1* nor *drp-1* altered lipid droplets in HT115 and OP50. One explanation is that our microscopy-based assay is not sufficient to detect minor changes in the gut. Alternatively, and more likely, diet-derived lipids are probably the main components of DHS-3-positive structures (lipid droplets), while mitochondria-derived membranes are re-distributed across organelles.

(13) Fig. 4. It would be more convincing to test whether mitophagy, instead of autophagy, is also higher in *wah-1* mutant animals on HT115 and declined upon *lrk-1* and *drp-1* RNAi treatment.

We agree with the reviewer that an assessment of mitophagy would provide a more convincing understanding. However, available reporter lines for mitophagy are not as good as those for autophagy. We have tried to correlate mitochondrial signals with GFP::LGG-1-positive structures, however we are not confident about the data.

To strengthen our evidence and address the reviewer's request, we grew *wah-1* mutants on RNAi bacteria against *pink-1*, thereby regulating at least one of the pathways that controls mitophagy. Remarkably, we found that *pink-1* downregulation partially rescued the lifespan reduction and mitochondrial network defects of *wah-1* animals grown on HT115 bacteria. These exciting findings are

perfectly in line with our idea that high turnover of mitochondria and sustained flux of mitochondrial membranes toward other organelles may become increasingly difficult without an adequate lipid intake to replenish those diverted to other cellular structures or use to generate ATP. Consequently, prolonged autophagy and mitophagy can undermine mitochondrial network homeostasis, potentially triggering vicious cycles of degradation and dysfunction that further impair the cell's bioenergetic capacity. These data (Figure 4H-I) have been included and discussed within the framework of our current knowledge in the field.

We are aware that these new data raise further questions about signaling pathways and molecular adaptors that may be involved in the regulation of the mitochondrial network in long-lived vs short lived-mitochondrial mutants. Although additional work can always be added to a paper, we believe that future independent studies may be more suitable to complete the current picture by building up scientific knowledge on our current evidence. We hope that the reviewer agrees.

(14) Fig 4E shows that DRP-1 expression is induced in *wah-1* mutant animals particularly under the HT115 condition, which might be a consequence of increased mitochondrial stress. Therefore, the title "LRK-1 and DRP-1 sense lipid availability" seems not precise enough, because they are not direct sensors of lipids.

We revised the previous title as follows: "Diets modify *wah-1/AIFM1*-associated phenotypes via LRK-1 and DRP-1 expression". We remain open to modifying it again in accordance with the recommendations of the reviewers and the guidance provided by the Editor.

(15) According to the model proposed by authors (line 305), lower TAG levels from HT115 diet may exacerbate the energy crisis due to WAH-1 deficiency. This can be tested by measuring ATP levels in control and mutant worms fed with OP50, HT115, or $\Delta dgkA$.

ATP measurement in nematodes is always tricky, especially if it is based on commercial kits previously validated in mammalian cells. To overcome this limitation and encouraged by the reviewer's request, we decided to employ an integrated transgene expressing the ratiometric ATP indicator Queen-2m. Based on our confocal imaging analyses across experimental conditions, we found that *wah-1* mutants grew on either OP50 or K-12 $\Delta dgkA$ bacteria have higher Queen-2m signals in the muscle cells. These exciting new data suggest that diets can enhance cytosolic ATP levels (Figure 2I-J and 3K), despite the inherited WAH-1/AIF mutation.

Minor points

(16) Fig. 1J. *wah-1* mutant animals have reduced OCR. Do they have increased glycolysis to maintain their energy supply?

As highlighted in our manuscript, prominent proteomic signatures included the upregulation of enzymes involved in glucose metabolism (ALDO-1, GPD-3, ENOL-1, PYC-1, LDH-1, MDH-2) (Figure 1E and Supplementary figure S1G). Since most of the key regulator of glycolysis are consistently upregulated, we believe that these data suggest that glycolytic processes contribute to energy supply.

During the revision, we tried to study the lifespan of *wah-1* mutants grown on RNAi targeting glycolytic enzymes. However, we observed that *wah-1* mutants were highly sensitive to this treatment, resulting in ~60% of the animals being censored within the first 10-12 days (Supplementary panels for the reviewers, B). We have included these data for the reviewer's consideration, but we do not feel confident publishing them, as their robustness does not reach our standards.

(17) Fig. 2H. This result is rather distractive and dispensable to major conclusions. It may be moved to the supplementary figure. Moreover, the images are not convincing to support that *wah-1* is mainly expressed in the intestine "compared to other tissues". It is possible that this gene is also highly expressed elsewhere, but the signals are just masked by the bulky intestine tissue. The authors may check published single cell RNAseq databases for this gene expression pattern to confirm this point.

We have removed the panel as suggested by the reviewer.

(18) The title is a bit confusing and shall be modified to reflect major conclusions. So far the mechanism with "LRK-1 and DRP-1" is not clear enough, but dietary lipids is a very convincing point.

As mentioned earlier, we do agree to revise the title in accordance with the suggestions provided by both the reviewer and the Editor.

Reviewer #3 (Remarks to the Author):

The authors provided several lines of evidence to suggest that LRK-1 and DRP-1 act in the same pathway to sense diet-derived lipids and modify *wah-1*/AIFM1-associated disease phenotypes. This finding is interesting and will have impact in the field.

We would like to thank the reviewer for his/her kind words.

(19) Several links are still required to complete this finding. For example, the manuscript did not present sufficient data to support the relationship between diet-derived lipids and LRK-1 and DRP-1. The data related to LRK-1 and DRP-1 should also be verified in *wah-1(bon89)* fed OP50.

To further strength our evidence, we have carried out additional epistatic analyses. In this regard, we developed two *lrk-1 loss of function* alleles and a new line overexpressing DRP-1 in postmitotic cells. These experiments using the two different diets (HT115 and OP50) have been reported in Figure 5 and Supplementary figure S4. We hope that our additional findings are sufficient to address the reviewer's comment and further support our conclusions.

(20) Furthermore, *wah-1(bon89)* nematodes fed OP50 and HT115 had less green fluorescence than WT in Figure 2, indicating lower lipid content in *wah-1(bon89)*, but they exhibit different longevity changes compared to WT.

We are not sure how to interpret the reviewer's comment. If he/she refers to Figure 2I (now 3A), the lower DHS-3::GFP signal correlates with the reduced lipid accumulation in the gut, possibly because of an imbalance between fatty acid synthesis (as shown by lower FAT-7 expression) and beta oxidation. To address the importance of lipid availability to *wah-1* mutant survival, we employed two additional bacteria clones (BW25113, control and $\Delta dgkA$ mutant) from the Keio library. Having quantified their metabolic profiles and their effects on *C. elegans* lipid contents (Figure 3D), we assessed lipid droplets and lifespan of control and *wah-1* mutants on these two bacteria strains. Consistent with our data in OP50 and HT115 bacteria, lipid-rich bacteria (BW25113 $\Delta dgkA$ mutant) promoted lipid droplet accumulation in *wah-1* mutants and stimulated animal survival (Figure 3G-H). We hope that this clarification adequately addresses the reviewer's concern.

REFERENCES

- Bano D, Prehn JHM (2018) Apoptosis-Inducing Factor (AIF) in Physiology and Disease: The Tale of a Repented Natural Born Killer. *EBioMedicine* 30: 29-37
- Bertan F, Wischhof L, Scifo E, Guranda M, Jackson J, Marsal-Cots A, Piazzesi A, Stork M, Peitz M, Prehn JHM *et al* (2021) Comparative analysis of CI- and CIV-containing respiratory supercomplexes at single-cell resolution. *Cell Rep Methods* 1: 100002
- Brooks KK, Liang B, Watts JL (2009) The influence of bacterial diet on fat storage in *C. elegans*. *PLoS One* 4: e7545
- Harhai M, Foged MM, Zarges C, Landoni JC, Chollet S, Simonelli M, Recazens E, Lisci M, Laban N, Manley S *et al* (2025) An updated inventory of genes essential for oxidative phosphorylation identifies a mitochondrial origin in familial Meniere's disease. *Cell reports* 44: 116069
- Reinhardt C, Arena G, Nedara K, Edwards R, Brenner C, Tokatlidis K, Modjtahedi N (2020) AIF meets the CHCHD4/Mia40-dependent mitochondrial import pathway. *Biochim Biophys Acta Mol Basis Dis* 1866: 165746
- Rothemann RA, Pavlenko E, Mondal M, Gerlich S, Grobushkin P, Mostert S, Racho J, Weiss K, Stobbe D, Stillger K *et al* (2025) Interaction with AK2A links AIFM1 to cellular energy metabolism. *Mol Cell* 85: 2550-2566 e2556
- Stuhr NL, Curran SP (2020) Bacterial diets differentially alter lifespan and healthspan trajectories in *C. elegans*. *Commun Biol* 3: 653

Troulinaki K, Buttner S, Marsal Cots A, Maida S, Meyer K, Bertan F, Gioran A, Piazzesi A, Fornarelli A, Nicotera P *et al* (2018) WAH-1/AIF regulates mitochondrial oxidative phosphorylation in the nematode *Caenorhabditis elegans*. *Cell Death Discov* 4: 2

Watts JL, Ristow M (2017) Lipid and Carbohydrate Metabolism in *Caenorhabditis elegans*. *Genetics* 207: 413-446

Wischhof L, Gioran A, Sonntag-Bensch D, Piazzesi A, Stork M, Nicotera P, Bano D (2018) A disease-associated Aifm1 variant induces severe myopathy in knockin mice. *Mol Metab* 13: 10-23

Wischhof L, Scifo E, Ehninger D, Bano D (2022) AIFM1 beyond cell death: An overview of this OXPHOS-inducing factor in mitochondrial diseases. *EBioMedicine* 83: 104231

Supplementary panels for the reviewers

A

B